# Pathogenic variants in *SLF2* and *SMC5* cause segmented chromosomes and mosaic variegated hyperploidy

Embryonic development is dictated by tight regulation of DNA replication, cell division and differentiation. Mutations in DNA repair and replication genes disrupt this equilibrium, giving rise to neurodevelopmental disease characterized by microcephaly, short stature and chromosomal breakage. Here, we identify biallelic variants in two components of the RAD18-SLF1/2-SMC5/6 genome stability pathway, *SLF2* and *SMC5*, in 11 patients with microcephaly, short stature, cardiac abnormalities and anemia. Patient-derived cells exhibit a unique chromosomal instability phenotype consisting of segmented and dicentric chromosomes with mosaic variegated hyperploidy. To signify the importance of these segmented chromosomes, we have named this disorder Atelís (meaning - incomplete) Syndrome. Analysis of Atelís Syndrome cells reveals elevated levels of replication stress, partly due to a reduced ability to replicate through G-quadruplex DNA structures, and also loss of sister chromatid cohesion. Together, these data strengthen the functional link between SLF2 and the SMC5/6 complex, highlighting a distinct role for this pathway in maintaining genome stability.

Despite the fundamental nature of DNA replication and cell division, inherited variants in genes involved in these processes are an underlying cause of human disease. Whilst these syndromes usually display unique clinical features that define them diagnostically, they typically exhibit common neurodevelopmental deficits, such as severe microcephaly and pre- and post-natal growth retardation[1-3]. As such, many of these syndromes can be collectively referred to as microcephalic dwarfism (MD) disorders. This constellation of conditions includes Meier-Gorlin Syndrome, Seckel Syndrome Spectrum Disorders, Bloom Syndrome and Microcephalic Osteodysplastic Primordial Dwarfism type II and can be broadly classified as having deficiencies in one of three cellular processes: DNA replication, DNA repair, and mitotic cell division[1-4]. Although mechanistically distinct, the common clinical phenotypes exhibited by these diseases are thought to result from a reduction in cellular proliferation and/or excessive cell death in the developing embryo, which reduces the number of cells available to maintain normal foetal growth[5]. Cells from these patients often exhibit signs of increased genome instability, such as micronuclei and/or

elevated chromosome breakage. A distinct subgroup of these syndromes exhibit rare cytogenetic anomalies, for example, mosaic variegated aneuploidy syndrome (MVA)[6-8] caused by variants in the spindle assembly checkpoint genes *BUB1B*, *CEP57* and *TRIP13*, or railroad chromosomes and premature chromatid separation (PCS) associated with Warsaw Breakage Syndrome (WABS) and Cornelia de Lange syndrome, caused by variants in the helicase DDX11 and components of SMC1/3 cohesin complex respectively[9,10]. Whilst, the presence of these chromosomal abnormalities is a useful diagnostic tool they can also help dissect the cellular mechanisms underlying the disease pathology.

Here, we report 11 patients with a neurodevelopmental disorder overlapping clinically with MVA and Fanconi Anemia (FA) with pathogenic variants in SLF2 and SMC5, two components of the recently discovered RAD18-SLF1/2-SMC5/6 genome stability pathway[11]. The precise function of the SMC5/6 complex remains enigmatic, however, it has been linked to a number of fundamental processes, including DNA transcription, DNA replication, DNA repair and

✉ e-mail: lecaignec.c@chu-toulouse.fr; eridavis@luriechildrens.org; g.s.stewart@bham.ac.uk

chromosome segregation[12,13]. Evidence suggests that the primary function of this complex occurs during DNA replication to stabilize stalled forks, suppress the activity of pro-recombination factors and promote efficient replication through difficult-to-replicate and/or repetitive regions of the genome, such as rDNA and telomeres[14]. In contrast, the function of SLF1 and SLF2 remain unclear, other than a reported role in recruiting the SMC5/6 complex to sites of DNA damage[11].

Analysis of SLF2 and SMC5 patient-derived cell lines revealed spontaneous replication stress and multiple mitotic abnormalities that give rise to a unique, diagnostically relevant, genome instability phenotype consisting of segmented, dicentric and rail-road chromosomes, and mosaic variegated hyperploidy (MVH). The underlying basis for this chromosomal instability is not fully understood, but our data suggest that it may arise, in part, from the failed resolution of aberrant DNA structures during S-phase, such as G-quadruplexes (G4), potentially leading to a combination of under-replicated DNA and unresolved recombination intermediates persisting through to mitosis. Together, these data demonstrate that despite a hitherto unknown role as a core component of the SMC5/6 complex, SLF2 is essential for the SMC5/6 cohesin-like complex to maintain genome stability by regulating both DNA replication and cell division.

## Results

### Patients with microcephaly and short stature have biallelic *SLF2* (*FAM178A*) and *SMC5* variants

Whole exome sequencing (WES) was carried out on seven patients (P1, P2, P3, P4-1, P4-2, P5 and P6) from five families, presenting with microcephaly, short stature, mild to severe developmental delay and spontaneous chromosome breakage. After aligning WES reads to the reference genome, variant calling, and filtering for rare variants (MAF <0.005), analysis under a recessive model of inheritance identified biallelic variants in *SLF2* (*FAM178A*) in all seven patients. All identified *SLF2* variants segregated amongst family members (with the exception of patients P1 and P5 where parental material was unavailable) and were present at a frequency of <0.5% in the gnomAD database (Fig. 1a, c; Supplementary Data 1–7; Supplementary Fig. 1a). Comparative genomic hybridization (CGH) array analysis carried out on gDNA from patient P5 confirmed the homozygosity of the identified *SLF2* variant.

Given that SLF2 had been identified previously as part of the RAD18-SLF1/2-SMC5/6 genome stability pathway[11], we hypothesized that variants in other components of this pathway may also give rise to a similar neurodevelopmental disorder. By querying gene matching platforms, four patients exhibiting microcephaly and growth retardation that had undergone WES were identified to carry biallelic variants in *SMC5*: patient P7 (c.1110_1112del; p.Arg372del, c.1273C>T; p.Arg425Ter) and patients P8, P9-1 and P9-2 (c.2970C>G; p.His990Asp) (Fig. 1a, c; Supplementary Data 1; Supplementary Data 8–10; Supplementary Fig. 1b). All variants were verified by Sanger sequencing, segregated amongst family members in an autosomal recessive paradigm and were present at a frequency of <0.5% in gnomAD.

### *SLF2* and *SMC5* variants give rise to neurodevelopmental abnormalities, cardiac defects and anemia

All individuals with *SLF2* and *SMC5* variants presented with a similar clinical phenotype, including marked microcephaly (−3.57 to −11.88 SD) and a reduction in height (-2.19 to -8.24 SD) (Fig. 1b; Supplementary Data 1). Moreover, the majority of patients also exhibited a developmental delay along with learning difficulties. Mild skeletal defects (i.e., clinodactyly), skin hyperpigmentation and ocular abnormalities were present in several patients (Supplementary Data 1). Notably, two of seven SLF2 patients (P4-1, P5) and all four SMC5 patients (P7, P8, P9-1 and P9-2) displayed cardiac defects (Supplementary Data 1), such as atrial or ventricular defects, a phenotype

commonly observed in patients with cohesinopathies[15,16] but not DNA replication disorders. Furthermore, five of eleven patients (P3, P4-1, P4-2, P5, P9-2) also developed anemia, with one of these patients (P9-2) subsequently developing myelodysplastic syndrome (Supplementary Data 1). This, coupled with other clinical features, could potentially result in future cases being mistakenly diagnosed with an atypical form of FA in the absence of a clear genetic diagnosis using WES. This is particularly relevant since components of the SMC5/6 complex have been previously shown to functionally interact with the FA pathway to repair DNA damage[17]. Only one patient (P3) developed severe pulmonary disease similar to patients with variants in the SMC5/6 complex subunit NSMCE3[18,19], whereas insulin-resistant diabetes and metabolic dysfunction, which are characteristic to patients with *NSMCE2* variants were absent among this cohort[20]. Collectively, these clinical and genetic observations support the premise that variants in *SLF2* and *SMC5* cause microcephaly and short stature associated with cardiac defects and the development of anemia.

### *SLF2* and *SMC5* variants compromise protein stability, interactions with other components of the RAD18-SLF1/2-SMC5/6 pathway and recruitment to sites of DNA damage

To determine the pathogenicity of the identified patient variants, we carried out western blotting on extracts from SLF2 patient-derived cell lines (SLF2-P1, SLF2-P2, SLF2-P3 and SLF2-P4-1) to ascertain if SLF2 protein abundance or stability was compromised. Notably, all four of the SLF2-mutant patient cell lines examined exhibited a reduction or absence of detectable full length SLF2 protein whilst maintaining wild type (WT) levels of RAD18, SMC5, and SMC6 protein (Fig. 2a). SLF1 protein level was not tested due to the absence of an available antibody.

We next investigated the *SLF2* variants in patients P2 and P3 in more detail. Analysis of cDNA from the SLF2-P3 cell line demonstrated that the synonymous homozygous variant c.3330G>A (p.Arg1110Arg), disrupted splicing leading to an in-frame deletion of exon 17 (Supplementary Fig. 2a, b). We then analysed the impact of the c.3486G>C (p.Gln1162His) variant, present in patient P2, on splicing. Multiple *SLF2* transcripts are annotated in the human genome and although c.3486G>C (p.Gln1162His) introduces a nonsynonymous change in the two longest transcripts (NM_018121 and NM_001136123), it only affects mRNA splicing of the most abundant *SLF2* transcript (NM_018121) by impairing the exon 19 splice donor splice site (Supplementary Figs. 2c, 3a–e). The p.(Gln1162His) variant also displayed compromised protein stability when expressed transiently indicating that this variant disrupts both mRNA and protein stability (Supplementary Fig. 3f). Together, these data suggest that most of the identified *SLF2* variants have an adverse effect on protein stability.

In contrast, analysis of SMC5 patient cell lines revealed that the homozygous p.(His990Asp) variant present in patients P8, P9-1 and P9-2 had little detectable impact on the protein stability of SMC5, or RAD18, SLF2, and SMC6 (Fig. 2b). Only a cell line derived from patient P7 exhibited a reduced abundance of SMC5 protein, presumably due to the presence of a nonsense variant (p.Arg425Ter) on one of the *SMC5* alleles. As loss of Smc5 is embryonically lethal[21], it is possible that the *SMC5* variants are hypomorphic and that significant disruption of SMC5 protein stability to the extent observed with the SLF2 variants is incompatible with life.

SLF1 and SLF2 have been identified as bridging factors between RAD18 and the SMC5/6 complex at sites of stalled replication[11]. To address whether the SLF2 and SMC5 variants compromised their ability to bind components of the RAD18-SLF1/2-SMC5/6 pathway, we initially mapped the binding sites of RAD18, SLF1 and SMC6 on SLF2. Using co-immunoprecipitation analysis with tagged proteins, we determined that the binding of RAD18 and SLF1 to SLF2 requires the C-terminal 471 amino acids (aa702-1173), which also overlapped with the SMC6 binding site located at amino acids 589–810 (Supplementary

**a**

| Individual | Ancestry | Gene | Mutation 1 | Allele Frequency | Polyphen-2 score | Mutation 2 | Allele Frequency | Polyphen-2 score |
|---|---|---|---|---|---|---|---|---|
| P1 | UK | SLF2 | c.1006dup; p.(Arg336LysfsTer27) | - | NA | c.1006dup; p.(Arg336LysfsTer27) | - | NA |
| P2 | France | SLF2 | c.2444C>G; p.(Ser815Ter) | - | NA | c.3486G>C (ss); p.(Gln1162His) | - | 0.999 |
| P3 | Netherlands | SLF2 | c.3330G>A (ss) p.(Arg1110ArgΔexon17) | - | NA | c.3330G>A (ss); p.(Arg1110ArgΔexon17) | - | NA |
| P4-1 | Japan | SLF2 | c.2582A>T; p.(Asn861Ile) | 3.98e-6 | 0.996 | c.2719dup; p.(Ser907PhefsTer5) | - | NA |
| P4-2 | Japan | SLF2 | c.2582A>T; p.(Asn861Ile) | 3.98e-6 | 0.966 | c.2719dup; p.(Ser907PhefsTer5) | - | NA |
| P5 | Saudi Arabia | SLF2 | c.2347_2348del; p.(Asp783SerfsTer53) | - | NA | c.2347_2348del; p.(Asp783SerfsTer53) | - | NA |
| P6 | German | SLF2 | c.568C>T; p.(Arg190Ter) | 3.19e-5 | NA | c.568C>T; p.(Arg190Ter) | 3.19e-5 | NA |
| P7 | Spain | SMC5 | c.1110_1112del; p.(Arg372del) | - | NA | c.1273C>T; p.(Arg425Ter) | 3.99e-6 | NA |
| P8 | US | SMC5 | c.2970C>G; p.(His990Asp) | - | 1.000 | c.2970C>G; p.(His990Asp) | - | 1.000 |
| P9-1 | US | SMC5 | c.2970C>G; p.(His990Asp) | - | 1.000 | c.2970C>G; p.(His990Asp) | - | 1.000 |
| P9-2 | US | SMC5 | c.2970C>G; p.(His990Asp) | - | 1.000 | c.2970C>G; p.(His990Asp) | - | 1.000 |

**b**

**c**

**d**

Fig. 4a–d). All patient-associated variants in SLF2, with the exception of p.(Gln1162His), are located within or truncate the SLF1/RAD18 binding domain of SLF2 (Fig. 1c). Consistent with SLF1 binding being essential for SLF2 to mediate bridging between RAD18 and the SMC5/6 complex, co-immunoprecipitation studies using extracts from hydroxyurea (HU) treated SLF2 patient-derived LCLs revealed a failure of all

cell lines tested to co-purify SMC6 with RAD18 (Fig. 2c). Furthermore, all SLF2-mutant proteins, with the exception of p.(Gln1162His), failed to or exhibited a reduced ability to, be recruited to sites of DNA damage induced by laser micro-irradiation (Supplementary Fig. 4e).

We next extended the co-immunoprecipitation analysis to include SMC5 patient LCLs (Fig. 2d). The interaction between RAD18 and SMC6

**Fig. 1 | *SLF2* and *SMC5* variants cause severe microcephaly and short stature.** **a** Table listing biallelic *SLF2* and *SMC5* variants in 11 individuals. ss, splice site created or destroyed by variant. '−' denotes that the allele variant was not present in the gnomAD database. Scores predicting the pathogenicity of the identified missense variants in *SLF2* and *SMC5* were generated using Polyphen-2 (http://genetics.bwh.harvard.edu/pph2/). NA Not applicable. **b** Length and head circumference (occipital frontal circumference; OFC) at birth and at the age of last exam as z-scores (s.d. from population mean for age and sex; SD). Dashed line at −3 SD indicates cut-off for normal population distribution. Orange values indicate SMC5 patients and blue values indicate SLF2 patients. **c** Schematic of full length WT SLF2 protein and SLF2 patient variants. APIM, atypical PCNA binding motif. SMC, SMC5/6 binding region. SLF1, SLF1 binding region. **d** Schematic of full length WT SMC5 protein and SMC5 patient variants. CC coiled-coil region.

in SMC5-P8 and SMC5-P9-1 cells was observed to be at WT levels, suggesting that p.(His990Asp) had no discernible impact on the integrity of the RAD18-SLF1/2-SMC5/6 complex, whereas the association of RAD18 with SMC6 was partially affected in SMC5-P7 cells. However, both the p.(Arg372del) and p.(His990Asp) SMC5 mutants failed to re-localize efficiently to sites of laser micro-irradiation induced damage, with the latter being more severely affected (Supplementary Fig. 4f). These observations indicate that whilst these variants largely do not appear to compromise their binding to components of the RAD18-SLF1-SLF2-SMC5/6 pathway, they do affect their re-localization to and/or retention at sites of damage.

To gain insight into why the SMC5 mutants affected stability of the SMC5/6 complex at sites of damage, we carried out co-immunoprecipitation analysis to assess if these mutations affected binding to other components of the complex. Interestingly, whilst the p.(His990Asp) mutation did not significantly affect binding to other components of the SMC5/6 complex, the p.(Arg372del) significantly compromised binding to SLF2, SMC6 and NSMCE2 (Fig. 2e). Moreover, endogenous NSMCE2 exhibited reduced binding to SMC5 in cells from patient SMC5-P7 (Fig. 2f). Consistent with these observations, the Nse2 binding site on yeast Smc5 lies in close proximity to Lys368, which is the yeast functional equivalent of human SMC5 Arg372 (Supplementary Fig. 5). This suggests that the failure of the p.(Arg372del) mutant SMC5 to be recruited to sites of laser damage may be due to this mutation compromising the binding of other key components of the SMC5/6 complex.

To explore the possibility that the p.(His990Asp) may have a deleterious impact on the structure of the SMC5/6 complex, we compared the AlphaFold model for human SMC5 to the X-ray crystal structures for the head domain of *Pyrococcus furiosus* Rad50 (*Pf*.Rad50) in both the unliganded and ATP-bound forms[22]. Notably, His990 lies just upstream of the ATP-binding cassette (ABC) signature motif of Smc5 (Supplementary Fig. 6a), a region of the protein implicit in both binding ATP and mediating the complex set of conformational changes that occur when SMC proteins bind nucleotide[23]. Interestingly, His990 sits in a position functionally equivalent to Phe791 of *Pf*.Rad50 - a residue known to interact directly with the adenine moiety of bound ATP[22]. Whilst mutation of His990 to aspartic acid would appear to be tolerated and unlikely to cause any gross-misfolding of the protein, as judged by the lack of steric clashes produced by the mutation (Supplementary Fig. 6b), it removes an aromatic amino acid and replaces it with one carrying a negative charge. As such, this would alter the overall charge of a region that normally functions to accept the adenine moiety. Therefore, it is likely that the p.(His990Asp) mutation perturbs the ability of the complex to either bind or turnover ATP, in turn affecting its association with, or retention on chromatin[24].

**Cell cycle arrest and increased apoptosis in the developing brain underlies the development of microcephaly in zebrafish lacking *slf2* and *smc5***

To gain insight into how SLF2 and SMC5 patient-associated variants affect neurodevelopment, we utilized CRISPR-Cas9 genome editing to ablate the single zebrafish orthologs of each of *slf2* and *smc5* in zebrafish embryos. Single guide (sg) RNAs targeting the primary isoforms of *slf2* and *smc5* (Supplementary Fig. 7a, f) were injected, with or without recombinant Cas9 protein, into *-1.4col1a1:egfp* reporter embryos at the single-cell stage, which were allowed to develop until

3 days post-fertilization (dpf) (Supplementary Fig. 7b, c, g, h). This reporter allows visualization of craniofacial patterning during embryonal development[25]. Bright field lateral images were acquired to measure head size and ventral fluorescent images of GFP-positive cells allowed visualization of the pharyngeal skeleton. Similar to the clinical phenotype exhibited by SLF2 and SMC5 patients, zebrafish embryos lacking *slf2* and *smc5* displayed a significant reduction in head size and aberrant craniofacial patterning, as indicated by a broadening of the angle of the ceratohyal cartilage; a major mandibular structure (Fig. 3a–f). Furthermore, unlike *smc5*, which is an essential gene[21], we were able to generate stable F2 *slf2* mutants possessing a frameshifting 8 bp deletion allele in *slf2* exon 7 (c.515_522del; p.Ser172_Ser174fsTer191; Supplementary Fig. 7d, e). Consistent with our observations from F0 embryos injected with sgRNA and Cas9, stable F2 *slf2* null mutants also exhibited microcephaly and aberrant craniofacial patterning (Fig. 3g).

To validate these findings, we used morpholinos (MO) to suppress the expression of *slf2* and *smc5* in zebrafish embryos. Splice blocking MO targeting the single zebrafish ortholog of each gene, *slf2* (exon 11) and *smc5* (exon 3), were designed and depletion of *slf2* and *smc5* mRNA was confirmed by RT-PCR after injection into zebrafish larvae (Supplementary Fig. 8a–d). MO were injected into *-1.4col1a1:egfp* reporter embryos at the single-cell stage. Injected embryos were reared to 3 dpf and then bright field images were acquired to measure head size and ventral fluorescent images of GFP-positive cells to visualize the pharyngeal skeleton. Comparable to our observations from the zebrafish embryos lacking *slf2* and *smc5*, zebrafish embryos depleted of *slf2* and *smc5* using MO also displayed a significant reduction in head size and aberrant craniofacial patterning in the pharyngeal skeleton (Supplementary Figs. 8e–h, 9a–f), which could both be rescued by re-expression of WT human *SLF2* or *SMC5* mRNA.

To confirm the pathogenicity of the *SMC5* disease associated variants we utilized our *smc5* morphant zebrafish model to ascertain whether the three patient-associated *SMC5* variants could rescue the developmental abnormalities caused by loss of *smc5* expression. Neither the p.(Arg425Ter), p.(Arg372del) nor p.(His990Asp) variants could complement the reduced head size and increased ceratohyal angle resulting from *smc5* depletion (Supplementary Fig. 9g–i), reinforcing that they confer a loss of function effect. In contrast, both the head size and ceratohyal angle could be restored to normal following expression of WT human *SMC5* or a polymorphic *SMC5* variant, p.(Arg733Gln), identified from gnomAD.

To investigate the two principal underlying causes of microcephaly, slowed cell cycle progression and/or increased apoptosis in the developing brain[2,26–28], fixed whole-mount *slf2* and *smc5* depleted zebrafish embryos were stained with markers of cell cycle stage (G2/M: phospho-histone H3 serine-10) and apoptosis (TUNEL). F0 CRISPR embryos injected with either *slf2* or *smc5* sgRNA with recombinant Cas9 (Fig. 4) exhibited a pronounced increase in both phospho-histone H3 and TUNEL staining in the developing brain when compared to control zebrafish. Importantly, this phenotype was recapitulated in zebrafish embryos transfected with *slf2* or *smc5* MO, which could be complemented by re-expression of the orthologous WT human mRNA (Supplementary Fig. 10). Together, these in vivo data confirm that a functional RAD18-SLF1/2-SMC5/6 pathway is required for normal development of the brain and cartilaginous structures, and compromising this pathway triggers a G2/M cell cycle arrest and the onset of apoptosis leading to microcephaly.

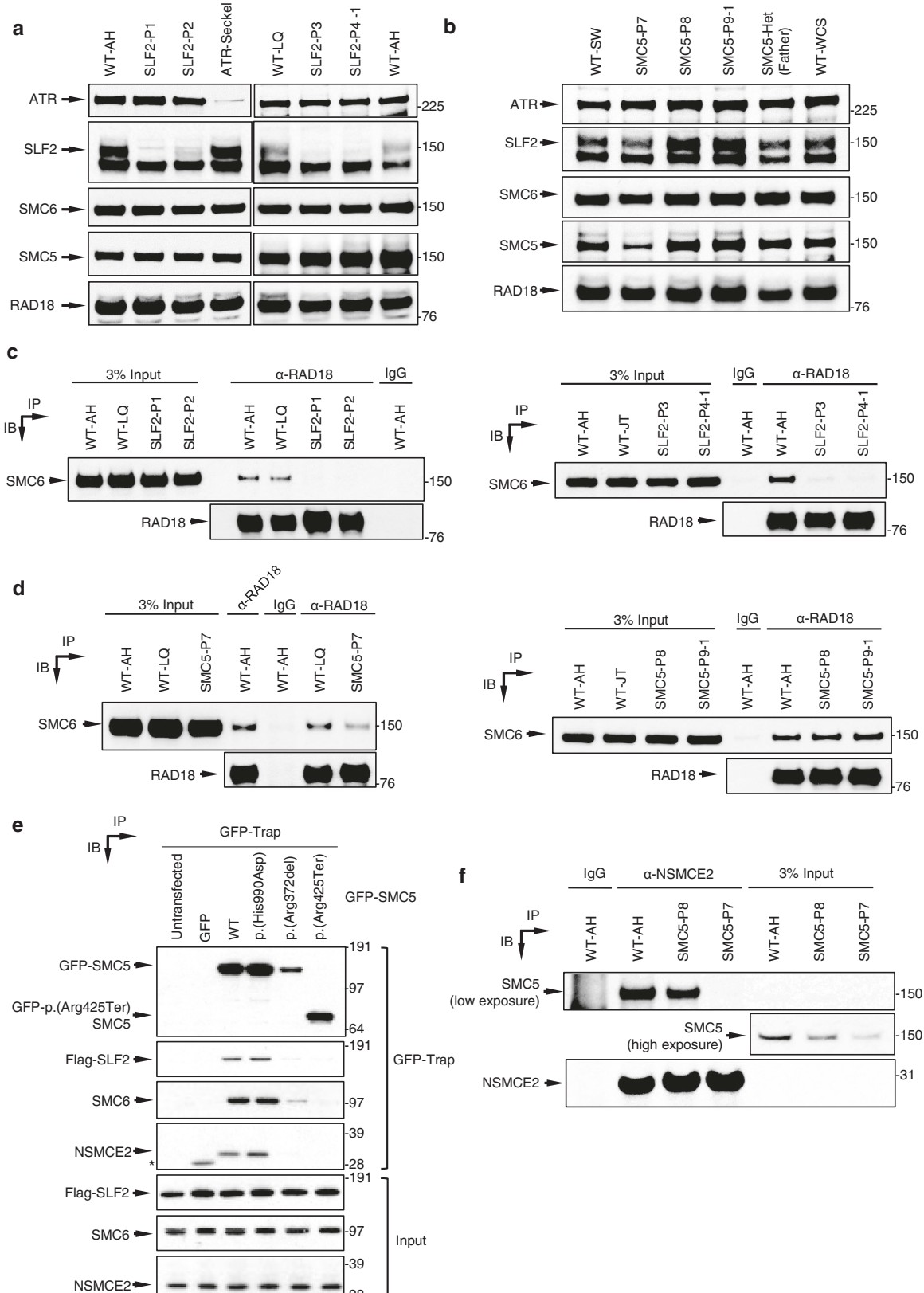

## SLF2/SMC5 mutant patient-derived cell lines exhibit increased spontaneous replication stress

Although the SMC5/6 complex has been implicated in regulating numerous DNA repair and replication pathways, it is thought that its primary function is to promote efficient replication[14,29]. Therefore, we used DNA fiber analysis to study the impact of SLF2 and SMC5 variants

on replication dynamics. All *SLF2* and *SMC5* mutant LCLs examined exhibited a significant increase in spontaneous replication fork stalling and fork asymmetry comparable to that observed in an LCL derived from an ATR-Seckel Syndrome patient (Fig. 5a–d). Importantly, this increased spontaneous replication fork stalling was also observed in patient-derived fibroblasts and could be suppressed by re-expressing

**Fig. 2 | Impact of patient-associated variants on the stability of SLF2 and SMC5 protein and the integrity of the SMC5/6 complex. a** Representative immunoblot analysis of cell extracts from lymphoblastoid (LCL) cell lines derived from patients with variants in *SLF2*. WT-AH and WT-LQ (WT wild type) indicate unrelated heathy individuals. **b** Representative immunoblot analysis of cell extracts from LCLs derived from patients with variants in *SMC5*. WT-SW and WT-WCS indicate unrelated heathy individuals. **c, d** Whole-cell extracts prepared from WT cell lines, SLF2 patient LCLs (**c**) or SMC5 patient LCLs (**d**) were subjected to immunoprecipitation with the indicated antibodies, and inputs and immunoprecipitates (IP) were analysed by immunoblotting (IB). **e** U-2 OS cells expressing Flag-SLF2 were

transfected with WT or mutant GFP-SMC5. GFP-SMC5 was precipitated from cell extracts using GFP-Trap beads and co-precipitated proteins were detected using immunoblotting with the indicated antibodies. *represents a cross-reaction of the NSMCE2 antibody to GFP. **f** Whole-cell extracts prepared from WT cell lines or SMC5 patient LCLs were subjected to immunoprecipitation with the indicated antibody, and inputs and immunoprecipitates were analysed by immunoblotting. Immunoblotting and immunoprecipitation experiments in (**a, b, c, d, f**) are representative of two independent experiments with similar results. Panel e is representative of three independent experiments with similar results.

---

WT *SLF2* or *SMC5* (Fig. 5e, f; Supplementary Fig. 11a, b). Unlike the ATR-Seckel cell line, all the SLF2-mutant LCLs and one of the SMC5 mutant LCLs exhibited WT levels of replication fork speed (Supplementary Fig. 11c, d). In contrast, LCLs carrying the homozygous p.(His990Asp) exhibited a moderate reduction in replication fork speed.

To confirm these observations, we used CRISPR-Cas9 gene editing in U-2 OS cells to generate SLF2 knockout clones. Despite several attempts we were unable to generate complete SLF2 knockout clones. Rather, we generated two hypomorphic (HM) clones, each with one expressed mutant allele of *SLF2* in conjunction with one or more truncating mutant alleles: SLF2 HM cl.1 (p.Asn411Lysins16, p.Ser403-Ter, p.Asn411LysfsTer3) and SLF2 HM cl.2 (p.Asp398_Ser404del, p.Ser403ThrfsTer14). These clones were subsequently complemented by re-expressing WT *SLF2* (Supplementary Fig. 12). Importantly, DNA fiber analysis of these SLF2 HM clones demonstrated that the vector complemented SLF2 HM cell lines exhibited significantly elevated levels of spontaneous fork stalling compared to the WT SLF2 complemented clones (Fig. 5g).

Since spontaneous replication stress exhibited by cells can be attributed to defective ATR-dependent DNA damage signaling, we used DNA fiber analysis and western blotting to monitor activation of the ATR-dependent stress response[30,31]. In contrast to the ATR-Seckel syndrome cell line, all the SLF2 or SMC5 patient cell lines were capable of activating ATR or the intra-S phase checkpoint in response to HU and MMC (Supplementary Figs. 11e, f, 13) indicating that dysregulation of the ATR stress response pathway does not account for the observed DNA replication defects. This is consistent with previous work demonstrating that loss of the SMC5/6 pathway does not affect activation of the ATR-dependent DDR[17].

We next investigated the cellular impact of the increased spontaneous replication fork instability observed in the patient cell lines using different markers of replication stress. Significantly, both SLF2 and SMC5 patient cell lines exhibited elevated signs of spontaneous replication stress including the presence of DNA double strand breaks (DSBs) in S-phase cells (53BP1 foci in EdU positive cells), an increased frequency of mitotic cells undergoing mitotic DNA synthesis (MiDAS), elevated levels of 53BP1 G1 bodies and the formation of micronuclei (Fig. 6a–d, Supplementary Fig. 14a–d)[17,29]. Crucially, all these phenotypes could be complemented by re-expressing either WT *SLF2* or *SMC5* (Fig. 6). Moreover, the U-2 OS SLF2 HM cell lines also exhibited elevated levels of micronuclei compared to the corrected WT SLF2 expressing clones (Fig. 6e).

### Hypomorphic variants in *SLF2* and *SMC5* are associated with mitotic abnormalities, segmented chromosomes, cohesion defects and mosaic variegated hyperploidy

Consistent with the elevated levels of spontaneous replication stress, LCLs derived from SLF2 and SMC5 mutant patients all exhibited increased levels of chromosomal aberrations (such as chromosome and chromatid gaps/breaks and chromosome radials) comparable to that observed in an ATR-Seckel Syndrome LCL (Fig. 6f, g). Notably, this phenotype was not significantly exacerbated by exposure to either APH or MMC, unlike LCLs from an ATR-Seckel Syndrome patient

(Supplementary Fig. 15a, b). Importantly, the elevated spontaneous levels of chromosomal aberrations in the SLF2/SMC5 patient fibroblasts and the U-2 OS SLF2 HM cells, were rescued by re-expression of either WT *SLF2* or *SMC5* (Fig. 6h, i).

In addition to the spontaneous chromosomal aberrations, metaphase spread analysis of both the peripheral blood and patient-derived LCLs of SLF2 and SMC5 patients revealed that a significant subset of cells exhibited large increases in chromosome numbers, with some metaphases having >100 chromosomes (Fig. 7a; Supplementary Figs. 16a, b, 17a). Unlike MVA, which typically involves the loss/gain of small numbers of chromosomes, the cytogenetic abnormality observed in SLF2 and SMC5 patient cells predominantly involved huge chromosomal gains. Therefore, we have termed this cytogenetic abnormality mosaic variegated hyperploidy (MVH), i.e., chromosome number >46.

To investigate the cause of the MVH, we explored whether SLF2 or SMC5 patient-derived cell lines exhibited spontaneous mitotic abnormalities. Both SLF2 and SMC5 patient fibroblast cell lines, and U-2 OS SLF2 HM cells, displayed a significant increase in mitotic cells with lagging chromosomes in empty vector complemented cells compared to cells re-expressing WT protein (Fig. 7b–d), consistent with previous reports[17,29,32]. Additionally, when we examined the origins of these lagging chromosomes/micronuclei using CENPA as a marker of centromeres, it was evident that a significant proportion of the micronuclei were positive for CENPA, suggesting that they could have resulted from failed mitotic segregation (Supplementary Fig. 16c, d). This is supportive of the RAD18-SLF1/2-SMC5/6 pathway playing an important role in promoting proper chromosomal segregation.

Since SMC5/6 forms a cohesin-like complex and has been implicated in facilitating centromeric and sister chromatid cohesion[21,32–35], we analysed metaphase spreads from SLF2 and SMC5 patient-derived cells for the presence of cohesion defects. SLF2 and SMC5 peripheral blood lymphocytes showed loss of sister chromatid cohesion as evidenced by the presence of rail-road chromosomes (Fig. 7e; Supplementary Fig. 17b). Moreover, SLF2 and SMC5 patient-derived LCLs exhibited PCS after treatment with the proteasome inhibitor MG132, which is known to induce cohesion fatigue by preventing the metaphase-to-anaphase transition[36] (Fig. 7f). Together, these observations suggest that the MVH characteristic to SLF2 and SMC5 patient cells may also be caused by PCS resulting from cohesion fatigue.

However, given the extent of the karyotypic abnormalities it seemed plausible that other cellular defects may contribute to the large increases in chromosome number seen in SLF2 and SMC5 mutant cell lines in addition to PCS. Replication stress can trigger centrosome amplification via fragmentation of the pericentriolar material (PCM)[37] or premature centriole disengagement, which can lead to mitotic arrest and aneuploidy-induced cell death and microcephaly[38]. To investigate whether centrosome abnormalities could contribute to the cellular pathology associated with SLF2 and SMC5 dysfunction, patient-derived cell lines were subjected to immunofluorescence with antibodies to PCNT1 (a component of the PCM) and mitosin/CENPF (marker of S/G2 cells) before and after incubation with aphidicolin

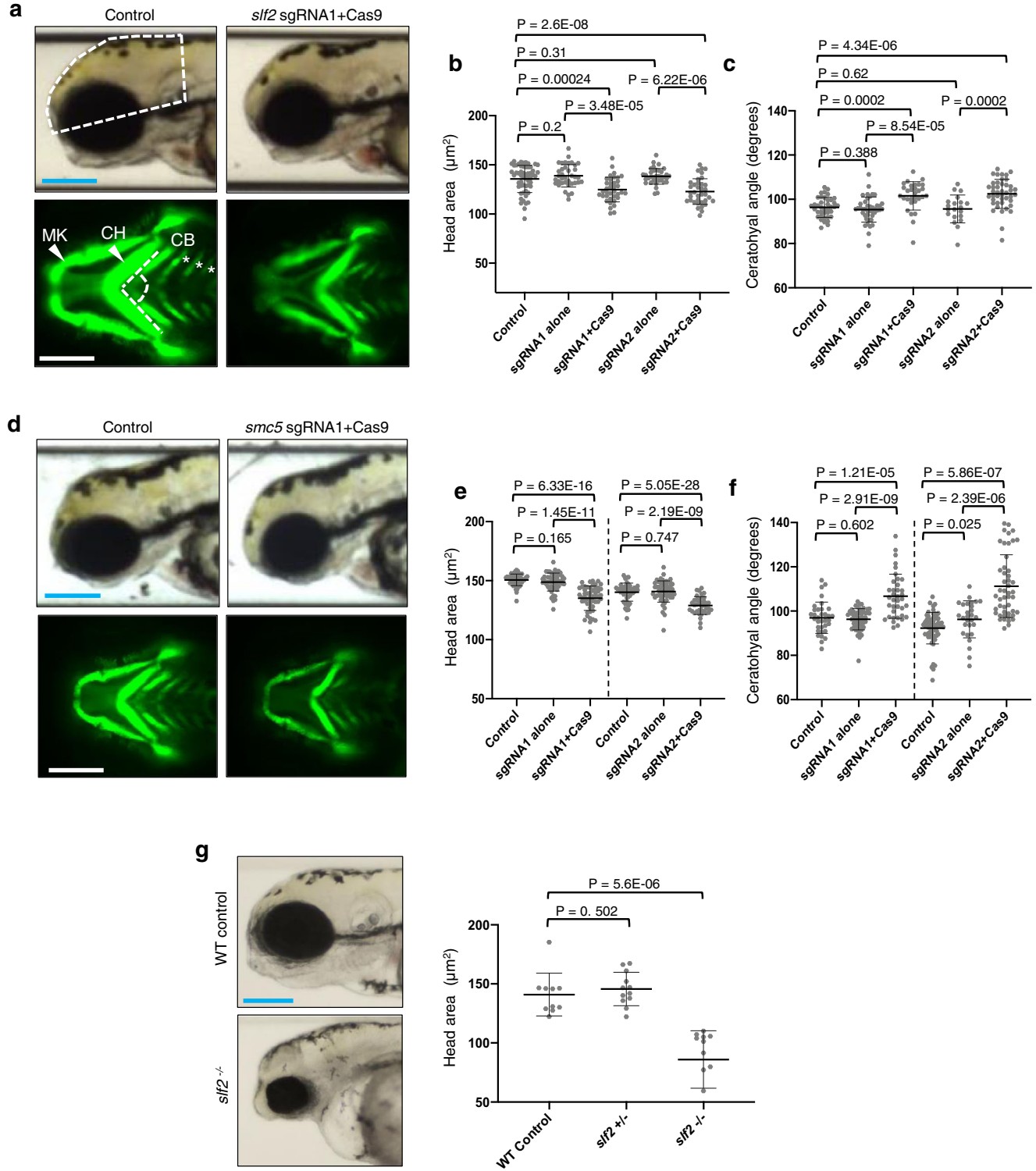

(APH). Notably, following APH exposure a significant proportion of S/ G2 cells possessed more than two centrosomes (Fig. 7g). We also observed that APH treatment had a profound effect on mitosis with >10–50% of SLF2 and SMC5 patient-derived LCLs exhibiting multi-polar spindles during mitosis (Fig. 7h, Supplementary Fig. 16e). This increase in centrosome number and multi-polar spindles is not due to higher levels of replication stress in the APH treated patient cells as quantification of APH-induced G1 53BP1 bodies revealed no difference between empty vector and WT SLF2/SMC5 complemented cells (Fig. 7i). Therefore, it is likely that the MVH observed in SLF2 and SMC5

patient cells arises as a consequence of multiple defects including unresolved replication stress, PCS, chromosome mis-segregation and centrosome amplification.

### SLF2/SMC5 mutant cells are unable to replicate efficiently in the presence of stabilized G-quadruplex structures

During our analysis of metaphase spreads of peripheral blood lym-phocytes from SLF2 and SMC5 patients, we noted that among the increased levels of spontaneous chromosomal damage, two distinct types of chromosome abnormality were evident (Fig. 8a;

**Fig. 3 | Loss of *slf2* and *smc5* in zebrafish give rise to microcephaly and aberrant craniofacial patterning. a** Top: Representative lateral bright field images acquired at 3 days post-fertilization (dpf); white dashed shape depicts head size measured. Bottom: Representative ventral images of GFP signal from the anterior region of *−1.4col1a1:egfp* transgenic reporter larvae at 3 dpf. The white dashed lines show the ceratohyal angle. **b** Quantification of lateral head size measurements. Larvae were injected with two independent sgRNAs targeting *slf2* with or without Cas9; *n* = 3 independent experiments (left to right; 56, 37, 37, 36, 36 larvae/batch). **c** Quantification of the ceratohyal angle. Larvae were injected with two independent *slf2* sgRNAs: *n* = 3 independent experiments (left to right; 39, 42, 30, 20, 44 larvae/ batch). **d** Top: Representative lateral bright field images at 3 dpf. Bottom: Representative ventral images of GFP signal in the anterior region of *−1.4col1a1:egfp smc5* sgRNA1 transgenic larvae at 3 dpf. **e** Quantification of lateral head size measurements in 3 dpf larvae (as shown in panel **a**); *n* = 3 independent experiments (left to right; 50, 50, 52, 46, 53, 38 larvae/batch). The chart shows two independent

experiments for sgRNA1 and sgRNA2 with a vertical line grouping independent controls with test conditions. **f** Quantification of the ceratohyal angle. Larvae were injected with two independent *smc5* sgRNAs: *n* = 3 independent experiments (left to right; 34, 53, 37, 62, 28, 48 larvae/batch). The chart shows two independent experiments for sgRNA1 and sgRNA2 with a vertical line grouping independent controls with test conditions. **g** Left: Representative lateral bright field images of WT control and *slf2^−/−^* mutants at 3 dpf. Right: Quantification of lateral head size measurements in 3 dpf WT control and *slf2^−/−^* mutant larvae (as shown in **a**); *n* = 3 independent experiments (left to right; 10, 12, 12 larvae/batch). In (**a**, **b**): (top left) white dashed shape depicts head size measured; (bottom left) white dashed lines show the ceratohyal angle measured. MK Meckel's cartilage, CH ceratohyal cartilage (indicated with arrowheads, respectively), and CB ceratobranchial arches (asterisks). Scale bars represent 300 µm, with equivalent sizing across panels. Error bars represent standard deviation of the mean. Statistical differences were determined with an unpaired Student's *t* test (two sided).

Supplementary Fig. 18). The first type of abnormal chromosome, which we termed segmented chromosomes, contained one or more chromosome gaps/breaks along the body of the chromosome (type 1). Type 1 segmented chromosomes with two or more gaps/breaks were particularly evident in SLF2-P1 and SLF2-P3, whilst most of the segmented chromosomes in SLF2-P2 and SMC5-P7 possessed one gap/ break. The second type of abnormal chromosomal structure resembled a dicentric chromosome, which was confirmed by the presence of two centromeres using centromere-specific FISH probes (type 2) (Fig. 8b).

The type 1 segmented chromosomes were reminiscent of the chromosomal abnormalities resulting from combined inactivation of GEN1 and either MUS81 or SLX4, suggesting that they may be caused by an inability to resolve recombination intermediates[39,40]. Accordingly, both SLF2 and SMC5 patient-derived cell lines exhibited elevated levels of recombination as indicated by increased levels of spontaneous RAD51 foci and sister chromatid exchanges (SCEs) in the patient-derived fibroblasts and LCLs respectively (Fig. 8c, Supplementary Figs. 19a, b, 15c, d). This is in line with previous work demonstrating a role for the SMC5/6 complex in resolving recombination intermediates[41–44]. We also observed an increased frequency of telomeric SCEs in SLF2-mutant LCLs (Supplementary Fig. 19c), which could, in part, contribute to the generation of the observed dicentric chromosomes. To investigate whether the spontaneous chromosomal aberrations observed in SLF2/SMC5 mutant cells could arise as a consequence of the presence of unresolved HR intermediates, we examined the effect of stably expressing the bacterial Holliday junction resolvase, RusA, in patient-derived cell lines on genome stability[40]. In line with SLF2 and SMC5 dysfunction causing unresolved HR intermediates to accumulate and this leading to increased genome instability, expression of WT RusA increased the level of spontaneous chromosome aberrations in SLF2/SMC5 mutant cells lines complemented with an empty vector but not with WT SLF2 or SMC5 (Supplementary Fig. 19d, e).

It is known that the SMC5/6 complex is important for the dissolution of replication stress-induced recombination, especially at repetitive regions prone to forming secondary structures and natural replication pause site intermediates[41,43–46]. This is consistent with our observations that the replication stress phenotype observed in SLF2/ SMC5 mutant cells was not markedly exacerbated by exposure to MMC, APH and HU (Fig. 5; Supplementary Figs. 11, 13). Recently, it has been shown that RNF168, which promotes the recruitment of the RAD18-SLF1/2-SMC5/6 pathway to damaged replication forks, is important for signaling the presence of G-quadruplex (G4) DNA structures stabilized by the RNA polymerase I inhibitor, CX5461[47]. Since cells deficient in BRCA1/2 and the cohesin-associated helicase DDX11 are also hypersensitive to this agent[48,49] and DDX11 was shown to function with SMC5/6 to repair DNA damage[17,50,51], we hypothesized that the RAD18-SLF1/2-SMC5/6 pathway might play a role in

suppressing replication stress at sites of stabilized G4 structures. To test this possibility, we first investigated the effects of CX5461 on DNA replication using DNA fiber analysis. This revealed that whilst WT SLF2 and SMC5 expressing patient fibroblasts could replicate normally in the presence of CX5461, SLF2 and SMC5 patient fibroblasts complemented with an empty vector exhibited a significant reduction in replication fork speed when incubated with this G4-stabilizing compound (Fig. 8d). Additionally, SLF2 and SMC5 patient-derived fibroblasts, LCLs and U-2 OS SLF2 HM cells treated with CX5461 exhibited increased levels of G1-phase 53BP1 bodies and chromosome aberrations (Fig. 8e, Supplementary Fig. 20a, c). In keeping with this, LCLs from SLF2-P1 and SMC5-P8 displayed an increased sensitivity to CX5461 (Fig. 8f). Strikingly, we also observed that CX5461 treatment induced a significant increase in the levels of type 1 segmented chromosomes in the SLF2 and SMC5 patient LCLs, but not in the WT LCLs (Supplementary Fig. 20b). These data suggest a role for SLF2 and the SMC5/6 complex in resolving replication stress at sites of stabilized G4 structures.

Whilst CX5461 is known to inhibit RNA polymerase I and stabilize G-quadruplexes, more recently it has also been identified as a TOP2 poison[52,53]. Given the pleiotropic nature of CX5461, we sought to identify which genotoxic lesion induced by CX5461 was causing the increased replication stress in cells deficient in components of the SMC5/6 complex. In this respect, we carried out DNA fiber and chromosomal aberration analysis on patient-derived cell lines following exposure to pyridostatin (a G-quadruplex stabilizer), etoposide (a TOP2 poison) and BMH21 (an RNA polymerase I inhibitor). Interestingly, only exposure to pyridostatin caused a significant reduction in replication progression and an increase in the levels of chromosome aberrations in SLF2 and SMC5 mutant cell lines (Fig. 8g, Supplementary Fig. 20d).

Taken together, these observations support the notion that the spontaneous replication stress and chromosomal instability displayed by cells from patients with SLF2/SMC5 mutations is caused, in part, by an inability to resolve a specific subset of replication-associated recombination intermediates arising at sites of G4 structures.

## Discussion

Disrupting the delicate balance between stem cell proliferation and differentiation profoundly affects embryonic development, particularly body growth and brain development. Rapidly proliferating pluripotent stem cells exhibit constitutively high levels of replication stress and as such are heavily reliant on replication-associated DNA damage response pathways to maintain genome stability. Unsurprisingly, patients with pathogenic variants in genes encoding components of the replisome, the DNA damage response (DDR) and factors that maintain sister chromatid cohesion exhibit developmental abnormalities including severe microcephaly and dwarfism. Furthermore, variants in centrosome components and regulators of the microtubule-spindle network can also result in these developmental abnormalities

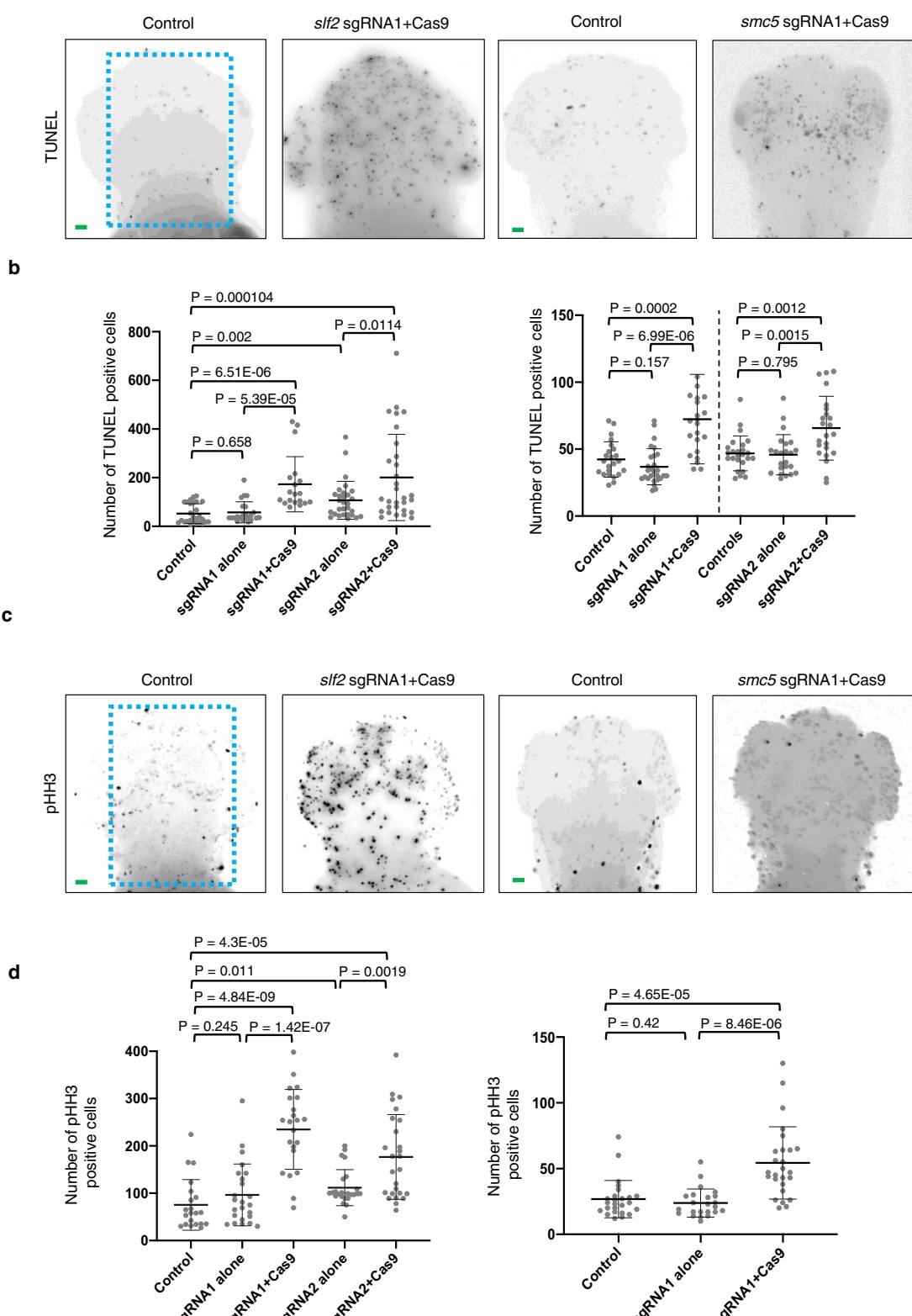

by affecting the orientation of the spindle pole and/or triggering excessive cell death through the generation of aneuploid cells[1]. However, it is often difficult to determine whether the cellular pathology underlying the development of these neurodevelopmental disorders results primarily from the presence of aberrant replication or defective mitosis[38,54,55].

Here we report the clinical and genetic characterization of 11 patients with biallelic variants in two components of the newly described RAD18-SLF1/2-SMC5/6 DDR pathway, *SLF2* and *SMC5*, exhibiting microcephaly, short stature, cardiac defects and anemia. However, in contrast to FA and other known disorders, cells from these patients exhibit a unique chromosomal instability phenotype,

**Fig. 4 | Loss of *slf2* and *smc5* induces apoptosis and altered cell cycle progression in zebrafish larvae. a** Representative dorsal inverted fluorescent images showing TUNEL positive cells in control and *slf2* F0 mutants at 2 dpf (left two panels), and control and *smc5* F0 mutants at 3 dpf (right two panels). The blue dashed line indicates the region of interest (ROI) quantified. Embryos of the same developmental stage and similar magnification were evaluated for all *slf2* and *smc5* conditions. **b** Left: Quantification of TUNEL positive cells in the ROI of control and *slf2* F0 mutants at 2 dpf shown in panel **a** (left to right; 27, 23, 19, 29, 30 embryos/condition were analysed from 3 independent experiments). Right: Quantification of TUNEL positive cells in control and *smc5* F0 mutants at 3 dpf in the ROI as shown in panel **a** (left to right; 37, 27, 22, 25, 23, 23 embryos/condition were analysed from 3 independent experiments). The chart shows two independent experiments for sgRNA1 and sgRNA2 with a vertical line grouping independent controls with test conditions. **c** Representative dorsal inverted fluorescent images showing phospho-histone H3 (pHH3) positive cells in control and *slf2* F0 mutants at 2 dpf (left two panels), and control and *slf2* F0 mutants at 3 dpf (right two panels). Embryos of the same developmental stage and similar magnification were evaluated for all *slf2* and *smc5* conditions. **d** Left: Quantification of pHH3 positive cells of control and *slf2* F0 mutants at 2 dpf in the ROI as shown in panel **a** (left to right; 21, 24, 22, 24, 26 embryos/condition were analysed from 3 independent experiments). Right: Quantification of pHH3 positive cells in the ROI in control and *smc5* F0 mutants at 3 dpf as shown in panel **a** (left to right; 25, 23, 26 embryos/condition were analysed from 3 independent experiments). For all panels: Statistical differences were determined with an unpaired Student's *t* test (two sided). Error bars represent standard deviation of the mean. Scale bars, 30 μm with equivalent sizing across panels.

hallmarked by segmented and dicentric chromosomes and mosaic variegated hyperploidy, arising from a combination of replication stress- and mitosis-associated cellular pathologies. Given that the segmented chromosomes seen in SLF2 and SMC5 patient cells represent a chromosome instability phenotype not previously associated with any known DNA repair or replication deficiency disorder, we have named this syndrome, Atelís Syndrome (ATS), after the Greek word for incomplete to signify the importance of these atelic or segmented chromosomes as a diagnostic marker of the disease.

The SMC5/6 complex has been shown to have many functions in the cell, including regulating homologous recombination (HR)-dependent DNA repair, stabilizing and restarting stalled replication forks, maintaining replication through highly repetitive regions of the genome, maintaining rDNA stability, elongating telomeres by ALT and controlling the topology of unusual DNA structures[12,14,56,57]. In contrast, little is known about the functions of SLF1 and SLF2, which were identified during a large proteomic screen of proteins associated with damaged replication forks[11]. However, it has been suggested that SLF1 and SLF2 are functional orthologs of the yeast Nse5 and Nse6 proteins, respectively, which are important for localizing the SMC5/6 complex to DNA damage and regulating its ATPase activity[11,58–60].

Pursuant to the role of the SLF1/2-SMC5/6 complex in maintaining replication fork stability, we demonstrate that cells from ATS patients exhibit elevated levels of spontaneous replication stress, although this was not exacerbated significantly following exposure to replication stress-inducing agents (HU, MMC or APH). This suggests that the clinical phenotype resulting from variants in *SLF2* and *SMC5* may not simply arise from elevated levels of replication stress, but rather from deficits with a subset of replication forks, such as those replicating through difficult-to-replicate regions of the genome or encountering specific types of endogenous DNA lesions. Consistent with this hypothesis, ATS cells fail to replicate efficiently in the presence of stabilized G4 structures and accumulate chromosomal damage, suggesting that the RAD18-SLF1/2-SMC5/6 pathway functions to resolve replication intermediates occurring at these lesions. Since G4 structures have been shown to be enriched at telomere repeat sequences[61], a defect in the ability to replicate through these lesions could result in genome instability at telomeres, potentially explaining the presence of dicentric chromosomes in ATS patient cells.

ATS patients exhibit overlapping clinical and cellular features with WABS patients, including microcephaly, growth restriction, skin hyperpigmentation, ocular abnormalities and heart defects. Moreover, cell lines derived from both ATS and WABS patients exhibit loss of sister chromatid cohesion and premature chromatid separation[49]. Interestingly, the loss of sister chromatid cohesion in WABS cell lines is exacerbated upon exposure to replication stress-inducing genotoxins, including G4 stabilizing agents[49]. Notably, cells from *Ddx11* null mice display loss of sister chromatid cohesion, chromosome segregation errors and aneuploidy, which has been shown to induce a G2/M cell cycle delay and apoptosis[62]. This suggests that a failure to resolve

specific endogenous DNA lesions, such as G4 structures, in ATS cells may directly compromise cohesion, or exacerbate a pre-existing cohesion defect, thus giving rise to chromosome segregation defects and aneuploidy that triggers cell death in highly proliferative tissues, such as the developing brain.

It is clear that the RAD18-SLF1/2-SMC5/6 pathway plays additional cellular roles beyond promoting replication through G4 lesions. In yeast, the smc5/6 complex restrains recombination at programmed fork pause sites, for example, in the rDNA locus[43,44,63] and, in mammalian cells, SMC5/6 is involved in suppressing HR at highly repetitive sequences, e.g., rDNA, centromeres and telomeres[14,63]. Consistent with this, ATS cells exhibit elevated levels of RAD51 foci in S-phase cells and spontaneous SCEs and tSCEs. Interestingly, segmented chromosomes have been observed in cells that have a combined defect in both the Holliday junction dissolution and resolution pathways[64], indicating that the gaps in the type 1 segmented chromosomes may result from a failure to dissolve/resolve recombination intermediates[41].

Cells from *NSMCE2* and *NSMCE3* mutant patients are not known to display segmented or dicentric chromosomes, and whilst *NSMCE3* patient-derived cells exhibit aneuploidy and structural chromosome abnormalities, hyperploidy to the extent seen in ATS cells was not reported[18,20]. This indicates that neither NSMCE2 nor NSMCE3 subunits are essential for this SMC5/6 function, or that the hypomorphic variants in these genes retain sufficient function to suppress these chromosomal phenotypes. Consistent with the latter scenario, *Nsmce2* transgenic mice lacking SUMO E3 ligase activity developed normally, whereas a complete loss of *Nsmce2* resulted in early embryonic lethality associated with chromosome segregation defects[65]. Notably MEFs derived from the *Nsmce2* knockout mice exhibited increased spontaneous replication stress and genome instability due to a failure to detangle recombination intermediates similar to ATS patient cell lines (e.g., elevated levels of BRCA1 foci, increased sister chromatid and telomeric SCEs and chromosomal segregation errors)[65] indicating that ATS represents a more severe form of SMC5/6 dysfunction.

Interestingly, the clinical phenotype exhibited by patients with variants in the SMC5/6 complex components *NSMCE2* and *NSMCE3* are different from each other, with the former being associated with microcephalic primordial dwarfism and insulin resistance[20] and the latter being associated with severe pulmonary disease and immunodeficiency[18,19]. It is unclear why these clinical presentations are different, especially as the cellular phenotype resulting from *NSMCE2* and *NSMCE3* variants are similar[18,20]. One possible important cellular difference between the two disorders is that the patient-associated missense variants in *NSMCE3* result in the destabilization of the SMC5/6 complex to a much greater extent than the nonsense variants present in *NSMCE2* patients[18,20]. It is notable that the clinical phenotype of ATS patients more closely resembles that of *NSMCE2* patients than *NSMCE3* patients, and like *NSMCE2* patient variants, *SLF2* and *SMC5* patient variants do not destabilize the SMC5/6 complex to any significant degree.

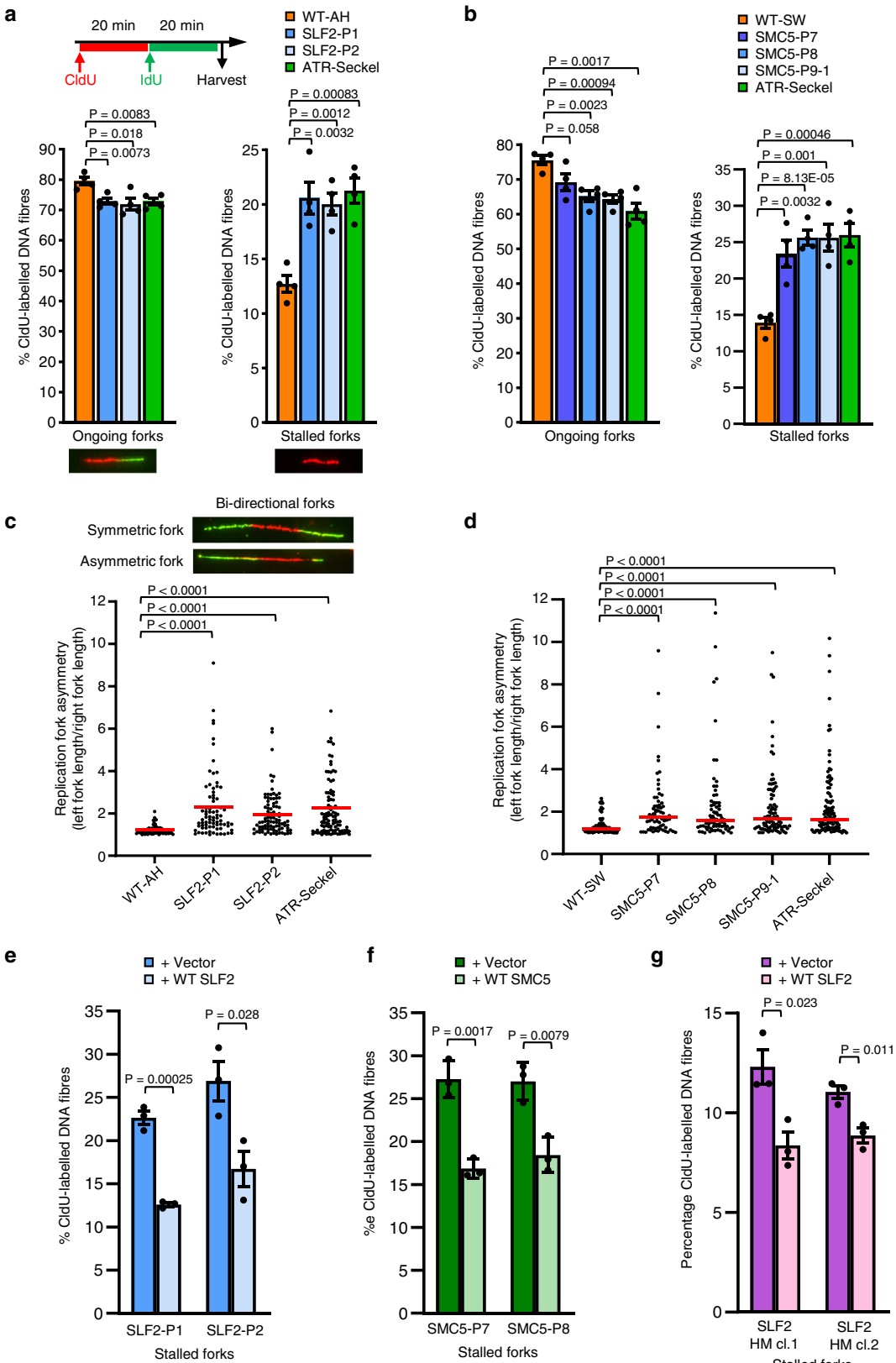

Taken together, we have demonstrated that variants in two components of the RAD18-SLF1/2-SMC5/6 pathway give rise to a FA/MVA-like disorder, termed Atelís Syndrome, with clinical and cellular features overlapping with WABS, MVA, NSMCE2 variants and FA. In vivo ablation of *slf2* and *smc5* in zebrafish recapitulate patient phenotypes including microcephaly and craniofacial patterning defects, likely due to concomitant cell cycle defects and apoptosis. We show that cells from ATS patients display a unique and complex chromosomal instability phenotype consisting of atelic (segmented) and dicentric chromosomes coupled with MVH, which should allow for cytogenetic diagnosis of patients with this disorder.

**Fig. 5 | Patient-derived cell lines from individuals with biallelic *SLF2* or *SMC5* variants exhibit increased levels of spontaneous replication fork instability.**
**a** Top: Schematic representation for DNA fiber analysis in untreated cells. The indicated cell lines were pulse-labeled with CldU for 20 min, then pulse-labeled with IdU for 20 min. Bottom: DNA fiber analysis of SLF2 patient-derived LCLs or LCLs from a WT individual. The percentage of ongoing forks (left) or stalled forks (right) was quantified. *n* = 4 independent experiments. A minimum of 1500 fork structures were counted. **b** DNA fiber analysis of SMC5 patient-derived LCLs or WT LCLs. Quantification of the levels of ongoing forks (left) or stalled forks (right). *n* = 4 independent experiments. A minimum of 750 fork structures were counted. **c**, **d** Quantification of replication fork asymmetry of WT, SLF2 patient (**c**) or SMC5 patient LCLs (**d**). *n* = 4 independent experiments. A minimum of 75 fork structures were counted. Red lines denote median values. A Mann-Whitney rank sum test was

performed for statistical analysis. Replication fork asymmetry represents the ratio of the left to right fork-track lengths of bidirectional replication forks. **e**, **f** DNA fiber analysis of SLF2 (**e**) and SMC5 (**f**) mutant fibroblast cell lines infected with lentiviruses encoding WT SLF2, WT SMC5, or an empty vector. The percentage of ongoing forks (left) or stalled forks (right) in untreated cells was quantified. A minimum of 350 fork structures in total were counted over 3 independent experiments. **g** DNA fiber analysis of U-2-OS SLF2 CRISPR hypomorphic (HM) cells infected with lentiviruses encoding WT SLF2 or an empty vector. The percentage of stalled forks in untreated cells was quantified. A minimum of 1000 fork structures in total were counted over 3 independent experiments. For (**a**, **b**, **e**, **f**, **g**); a Student's *t* test (two-sided, equal variance) was performed for statistical analysis and error bars denote SEM.

## Methods

### Research subjects
Informed consent was obtained from all participating families to take clinical samples and to publish clinical information in accordance with local approval regulations and in compliance with the Declaration of Helsinki principles. This study was approved by the West Midlands, Coventry and Warwickshire Research Ethics Committee (REC: 20/WM/0098), the Scottish Multicentre Research Ethics Committee (REC: 05/MRE00/74), the Lancaster General Hospital Institutional Review Board and the Institutional Review Boards of Yokohama City University Graduate School of Medicine (ID: A190800001) and Jichi Medical University (ID: G21-V06). A collaboration to study the pathological significance of the identified *SLF2* and *SMC5* variants was established via GeneMatcher[66].

### Exome sequencing
Genomic DNA from affected children and family members was extracted from peripheral blood using standard methods. Whole exome capture and sequencing was performed as described to a minimum of *30x* coverage[67]. Exome sequencing for families 8 and 9 was conducted in collaboration with the Regeneron Genetics Center as previously described[68]. Briefly, DNA was sheared (Covaris S2), exome capture performed using the Agilent SureSelect v5 enrichment kit according to manufacturer's instructions, and libraries were sequenced with 125 bp read-pairs using the Illumina HiSeq 2500 V4 platform. All analyses were performed as described[69]. Variants were confirmed by bidirectional capillary dye-terminator sequencing and annotated using the reference sequences, GenBank: NM_018121.4, NM_001136123.2 and NM_015110.4. Capillary sequencing was performed in the MRC Human Genetics Unit, Edinburgh, UK, the University of Birmingham, UK, the Bioscientia Institute for Medical Diagnostics, Germany, the Rare Disease Genomics Department, Yokohama City University Hospital, Japan and the Regeneron Genetics Center, Regeneron Pharmaceuticals Inc., USA.

### Cell lines
Patient-derived lymphoblastoid cell lines (LCLs) were generated from peripheral blood samples with Epstein Barr virus (EBV) transformation using standard methods and were maintained in RPMI-1640 medium (Life Technologies) supplemented with 10% FBS, L-glutamine and penicillin-streptomycin. The ATR-Seckel LCL used in this study was reported previously[31]. Dermal primary fibroblasts were grown from skin-punch biopsies and maintained in Dulbecco's modified Eagle's medium (DMEM; Thermo Fisher Scientific) supplemented with 20% FCS, 5% L-glutamine and 5% penicillin-streptomycin. Primary fibroblasts were immortalized with a lentivirus expressing human telomerase reverse transcriptase (hTERT) that was generated by transfecting 293FT cells (Thermo Fisher Scientific) with the plasmids: pLV-hTERT-IRES-hygro (Addgene #85140), psPax2 (Addgene #12260) and pMD2.G (Addgene #12259). Selection was performed using Hygromycin (Thermo Fisher Scientific) at 70 µg/ml. All LCLs were

routinely grown in RPMI-1640 (Thermo Fisher Scientific) supplemented with 10% FCS, 5% L-glutamine and 5% penicillin-streptomycin. Patient cell lines were validated using Sanger sequencing and immunoblotting. Fibroblast and U-2 OS cell complementation was carried out using the pLVX-IRES-Neo lentiviral vector (Takara Bio) encoding *2x*Myc-tagged *SLF2* or untagged *SMC5*.

293FT cells (Thermo Fisher Scientific) were maintained in DMEM supplemented with 10% FBS, 5% l-glutamine and 5% penicillin-streptomycin and U-2-OS cells were cultured in McCoy's 5A medium, supplemented with 10% FBS, and 5% penicillin/streptomycin. 293FT cells were transiently transfected with GFP-BLM or GFP expression vectors using Lipofectamine 2000 (Thermo Fisher Scientific). U-2 OS cells were transiently transfected with SLF2/SMC5 expression vectors using FuGENE 6 Transfection Reagent (E2692, Promega) or Lipofectamine 3000 Reagent (L3000015, Thermo Fisher Scientific) where indicated. Stable GFP-SMC5 cell lines were generated by G418 selection and low expressing clones were selected based on GFP expression. All cell lines were routinely tested for mycoplasma.

### Western blotting
Whole-cell extracts were obtained by sonication in UTB buffer (8 M urea, 50 mM Tris, 150 mM β-mercaptoethanol) and analyzed by SDS−PAGE following standard procedures. Protein samples were run on 6−12% acrylamide gels with SDS−PAGE and transferred onto a nitrocellulose membrane. Immunoblotting was performed using antibodies to: RAD18 (Fortis Life Sciences, A301-340A; 1:1000), SMC5 (Fortis Life Sciences, A300-236A; 1:500), SMC6 (Fortis Life Sciences, A300-237A; 1:2000), SLF2 (generated in house; 1:1000)[11], GAPDH (Genetex, GTX100118; 1:1000), Myc (Abcam, ab32; 1:1000), GFP (SCBT, sc-9996; 1:1000), HA (SCBT, sc-7392; 1:1000), α-Tubulin (Sigma-Aldrich, T9026; 1:20,000), ATR (Fortis Life Sciences, A300-137A; 1:1,000), phospho-ATR (Thr1989) (GeneTex, GTX128145; 1:500), FANCD2 (SCBT, sc-20022; 1:1,000), CHK1 (SCBT, sc-8408; 1:1,000), phospho-CHK1 (Ser345) (Cell Signaling Technology, 2341; 1:100), NBS1 (Genetex, GTX70224; 1:10,000); phospho-NBS1 (Ser343) (Abcam, 47272; 1:500); SMC1 (Fortis Life Sciences, A300-055A; 1:1,000); phospho-SMC1 (Ser966) (Fortis Life Sciences, A300-050A; 1:1,000); HA (Abcam, Ab9110; 1:1000). Loading controls for all blots were derived from re-probing the same membrane, except for the phospho-antibody immunoblots, for which paired gels were run simultaneously and blotted in parallel for phosphorylated and total proteins.

### Co-immunoprecipitation and GFP-Trap pull-downs
For GFP-Trap pulldown experiments with 293FT cells, cells transfected with plasmids using Lipofectamine 2000, were treated with 2 mM HU for 16 h and harvested. Cells were incubated in lysis buffer (150 mM NaCl, 50 mM Tris-HCl pH 7.5, 2 mM MgCl₂, 1% NP40, 90 U/ml Benzonase (Novagen) and EDTA-free protease inhibitor cocktail [Roche]) for 30 min with rotation at 4 °C. Cell lysates were then pre-cleared at 65,000 × *g* at 4 °C for 30 min. For GFP-Trap, 3−5 mg of lysate was

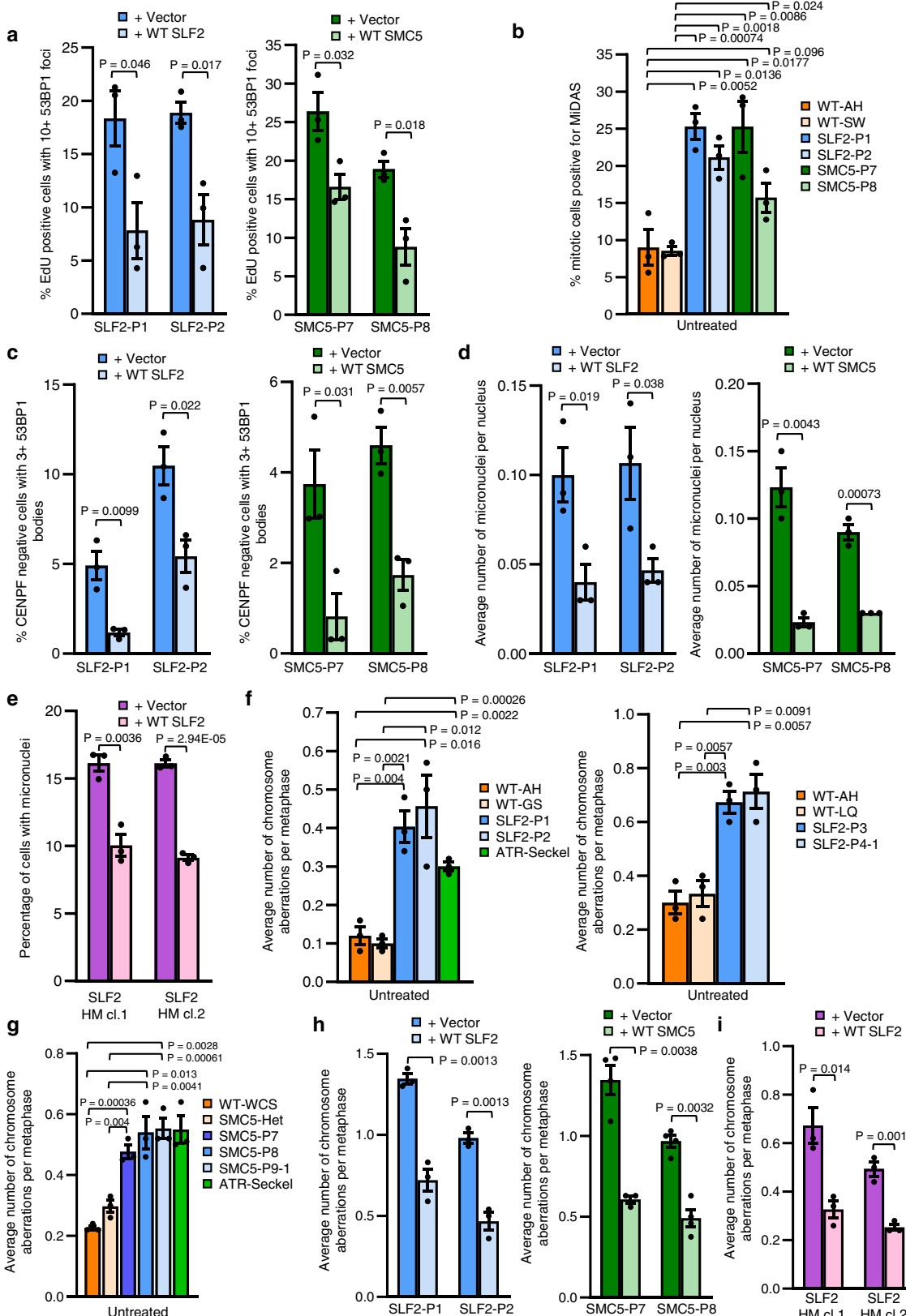

incubated with GFP-Trap agarose beads (ChromoTek) for 5 h at 4 °C. The resulting GFP-Trap complexes were washed with wash buffer (150 mM NaCl, 50 mM Tris-HCl pH 7.5, 0.5% NP40, and complete protease inhibitor cocktail [Roche]) and analysed by SDS−PAGE.

For immunoprecipitations from patient-derived LCLs, 3 mg of lysate (prepared with the same lysis buffer as above) was

immunoprecipitated with 5 μg of antibody (RAD18; Fortis Life Sciences, A301-340A or NSMCE2; Fortis Life Sciences, A304-129A) and protein A-sepharose beads (GE Healthcare). Complexes were washed with wash buffer (as described above) and analysed by SDS−PAGE. Experiments were carried out in the presence of Benzonase nuclease to exclude the possibility of interactions being mediated by DNA.

**Fig. 6 | SLF2 and SMC5 patient cells exhibit S-phase associated DNA damage.**
**a** Percentage of cells positive for EdU staining with >10 53BP1 foci in SLF2 and SMC5 mutant fibroblast cell lines infected with lentiviruses encoding WT SLF2, WT SMC5, or an empty vector. A minimum of 900 EdU positive cells across 3 independent experiments were counted. **b** SLF2 and SMC5 patient fibroblast cell lines were pulsed with 10 μM EdU for 45 min, fixed, and mitotic DNA synthesis was visualized by mitotic EdU incorporation following labeling with click chemistry. The percentage of mitotic cells with EdU foci was quantified. A minimum of 300 mitotic cells were counted. *n* = 3 independent experiments. **c** Immunofluorescent microscopy analysis to quantify the percentage of G1-phase cells (CENPF negative cells) with >3 53BP1 bodies in WT SLF2, WT SMC5, or an empty vector expressing SLF2 and SMC5 patient fibroblasts. *n* = 3 independent experiments. A minimum of 750 G1-phase cells were counted. **d** Levels of micronuclei in cells from (**c**). *n* = 3 independent experiments. A minimum of 2500 cells were counted. **e** Levels of micronuclei in U-2 OS SLF2 CRISPR HM cells infected with lentiviruses encoding WT SLF2 or an empty vector. *n* = 3 independent experiments. A minimum of 1700 cells were counted. **f, g** Quantification of the average number of chromosomal aberrations per metaphase (which includes chromatid/chromosome gaps, breaks, fragments and chromosomes radials) in WT, SLF2 patient (**f**), or SMC5 patient LCLs (**g**). *n* = 3 independent experiments. A minimum of 140 metaphases were counted. **h** Average number of chromosomal aberrations per metaphase (chromatid/chromosome gaps, breaks, fragments and chromosome radials) in SLF2 and SMC5 mutant fibroblast cell lines infected with lentiviruses encoding WT SLF2, WT SMC5, or an empty vector was quantified. *n* = 3 independent experiments. A minimum of 90 metaphases were counted. **i** Average number of chromosomal aberrations (chromatid/chromosome gaps, breaks, fragments and chromosome radials) per metaphase in U-2 OS SLF2 CRISPR HM cell lines expressing either WT SLF2 or an empty vector. *n* = 3 independent experiments. A minimum of 100 metaphases were counted. In all cases, a Student's *t* test (two-sided, equal variance) was performed for statistical analysis and error bars denote SEM.

For immunoprecipitations from U-2 OS cells, cell lysates were generated using EBC buffer (150-mM NaCl; 50-mM Tris, pH 7.5; 1-mM EDTA; 0.5% IGEPAL CA-630). Lysates were subject to Co-IP using Strep-Tactin Sepharose (IBA GmbH) prior to immunoblot using the following antibodies: GFP (sc-9996, SCBT; 1:1000), HA (sc-7392, SCBT; 1:1000), RAD18 (A301-340A, Fortis Life Sciences; 1:1000), SMC6 (A300-237A, Fortis Life Sciences; 1:2000), SMC5 (Fortis Life Sciences, A300-236A; 1:500), NSMCE2 (Fortis Life Sciences, A304-129A; 1:500), α-Tubulin (T9026, Sigma-Aldrich; 1:20000).

**Laser micro-irradiation**
U-2 OS cells were grown on coverslips and sensitized to laser induced DSB formation using 5-Bromo-2-deoxyuridine (B9285-50MG, Sigma-Aldrich) for 24 h. GFP-SLF2 expression vectors were transiently transfected 24 h prior and GFP-SMC5 stable expressing cells were used for micro-irradiation. Laser micro-irradiation induced DSB formation was performed as previously described[70] with 1 h allowed for recovery. Cells were pre-extracted using CSK buffer (100 mM NaCl, 10 mM HEPES, 3 mM MgCl$_2$, 300 mM Sucrose, 0.25% Triton-X-100, 1 mM PMSF) prior to fixation in formalin buffer (AMPQ43182, VWR) for 15 mins at room temperature (RT).

Fixed coverslips were blocked with 5% Bovine Serum Albumin (A7906, Sigma-Aldrich) for 1 h prior to staining with anti-γ-H2AX (Ser139) (1:1000, 05-636, Merck) and anti-GFP (1:500, PABG1, Chromotek) overnight at 4 °C. After PBS washes cells were stained with Alexa Fluor secondary antibodies and 4′,6-Diamidino-2-Phenylindole (DAPI, D1306, Molecular Probes) for 30 min at RT. After further washing, coverslips were dried completely and mounted for imaging using Mowiol (81381, Sigma-Aldrich).

**Zebrafish husbandry and embryo maintenance**
All zebrafish experiments were performed according to protocols approved by the Duke University and Northwestern University institutional animal care and use committees (IACUC). Wild type (WT: ZDR or NIH) adults or transgenic −1.4col1a1:egfp[25] adults were maintained on an AB background and subjected to natural matings to generate embryos for microinjection and/or phenotyping. Embryos were grown in egg water (0.3 g/L NaCl, 75 mg/L CaSO$_4$, 37.5 mg/L NaHCO$_3$, 0.003% methylene blue) at 28 °C until assessment. Zebrafish sex is unknown until animals are ~3 months old. Therefore, in the larvae at <5 days post fertilization, it is not possible to know how many males and females are present, and there should be no sex-dependent effects at this stage. However, adults that were used to generate embryos were crossed in a 1 male to 1 female ratio.

**CRISPR-Cas9 genome editing of zebrafish embryos**
Reciprocal translated BLAST of human *SLF2* (NP_060591.3) and *SMC5* (NP_055925.2) was performed against the zebrafish genome and found a single ortholog corresponding to either protein (transcripts targeted: *slf2*: ENSDART00000136689.3, *smc5*: ENSDART00000122170.4). To identify CRISPR/Cas9 single guide RNA (sgRNA) targets in both genes, CHOPCHOPv2[71] (and http://chopchop.cbu.uib.no) was used. sgRNAs were generated using the GeneArt precision gRNA synthesis kit (Thermo Fisher Scientific) according to the manufacturer's instructions (Supplementary Table 1). 1 nl of cocktail containing 100 pg sgRNA with or without 200 pg of Cas9 protein (PNA Bio) was injected into the cell of single-cell staged zebrafish embryos. To estimate the percentage mosaicism of genome-edited cells, genomic DNA from individual embryos was extracted at 2 days post fertilization (dpf; two controls and ten founder [F0] embryos per sgRNA). PCR was used to amplify the sgRNA targeted region using flanking primers and heteroduplex analysis was performed using polyacrylamide gel electrophoresis (PAGE). PCR products were denatured, reannealed slowly, and migrated on a 20% polyacrylamide gel (Thermo Fisher Scientific). PCR products from five embryos per sgRNA were randomly selected from the heteroduplex analysis, cloned into a TOPO-TA vector (Thermo Fisher Scientific) and sequenced using BigDye terminator 3.1 chemistry (Applied Biosystems). To isolate stable *slf2* mutants, F0 animals were crossed to WT ZDR adults and heterozygous F1 mutants bearing the c.515_522del (p.Ser172_Ser174fs191Ter) variant were identified. Mutant F1 adult siblings were inter-crossed to generate homozygous F2 animals for phenotyping. *slf2* mRNA expression level was monitored by qRT-PCR (QuantStudio, Thermo Fisher Scientific) using SYBR Green detection kit (Thermo Fisher Scientific) with normalization to *β-actin*.

**Transient suppression of *slf2* and *smc5* in zebrafish embryos**
Splice blocking morpholinos (MOs) were designed to target the *slf2* exon 11 (e11i11) and *smc5* exon 3 (e3i3) splice donor sites (Gene Tools; Supplementary Table 1)). Each gene was transiently suppressed independently by injecting 1 nl at different doses (3, 6, and 9 ng) into one to four cell staged zebrafish embryos. To validate MO efficiency, total RNA was extracted from pools of 2 dpf embryos (25 animals/condition; controls and MO-injected) using Trizol (Thermo Fisher Scientific) according to manufacturer's instructions. cDNA was synthesized with the QuantiTect Reverse Transcription kit (Qiagen), RT-PCR of the MO target locus was performed, and PCR products were separated on a 1% agarose gel. Resulting PCR bands were gel purified with the QIAquick gel extraction kit (Qiagen) and cloned into the TOPO-TA cloning vector (Thermo Fisher Scientific). Purified plasmids from resulting colonies (*n* = 4/PCR product) were sequenced using BigDye 3.1 terminator chemistry according to standard protocols.

**Molecular cloning and site-directed mutagenesis of human *SLF2* and *SMC5* constructs for expression of human proteins in zebrafish**
Full length Gateway-compatible *SLF2* (NM_018121.4) and *SMC5* (NM_015110.4) open reading frame (ORF) entry vectors were obtained (Horizon). WT ORFs of both genes were inserted into a pCS2+ Gateway

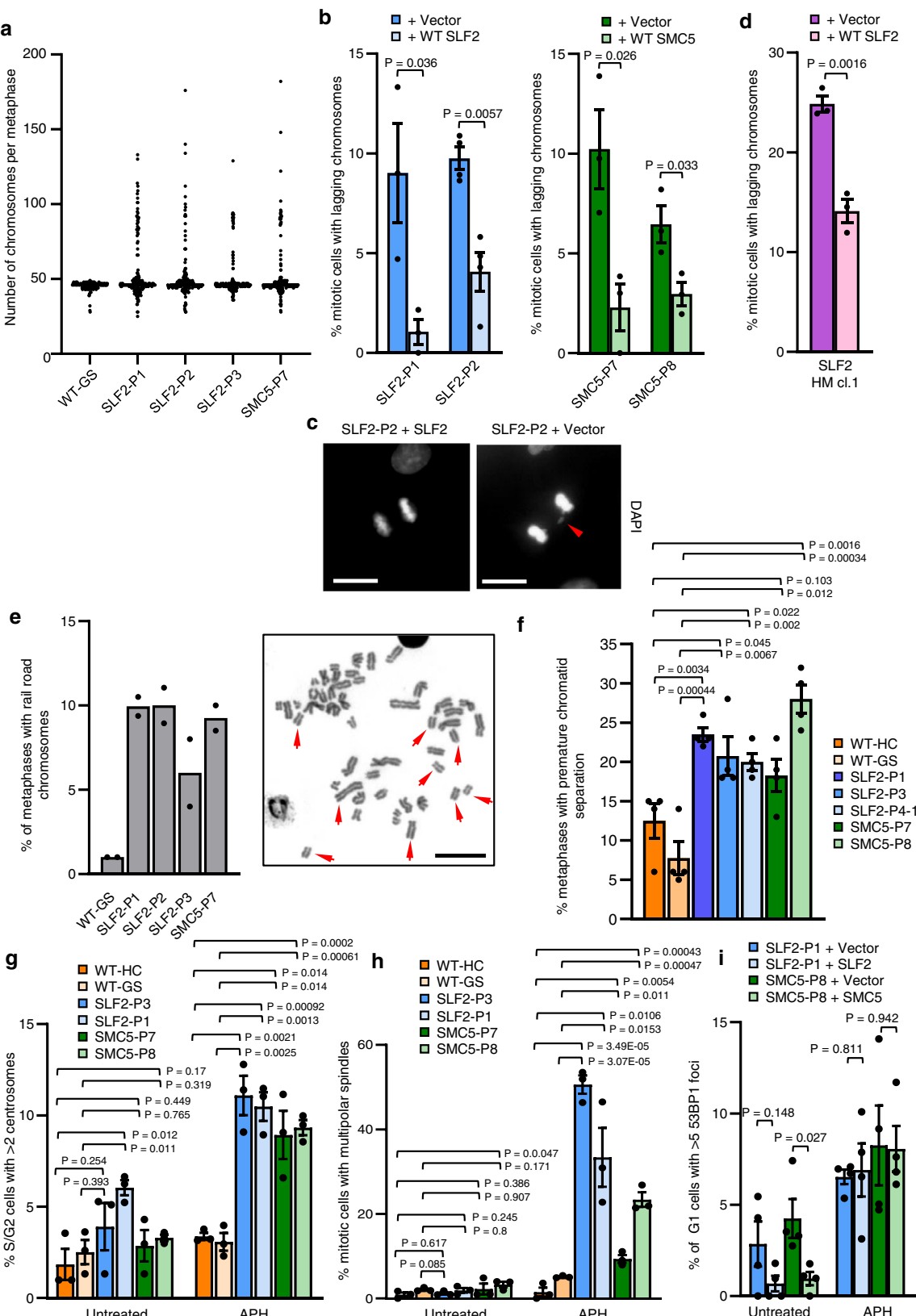

destination vector using LR clonase II (Thermo Fisher Scientific). *SMC5* variants identified in either affected individuals (p.His990Asp, p.Arg372del, p.Arg425Ter) or in gnomAD (dbSNP ID: rs59648118, p.(Arg733Gln); 16 homozygotes of 140,814 individuals, negative control) were inserted using site-directed mutagenesis as described (Supplementary Table 1)[72]. After full ORF sequence confirmation of all

WT and mutant plasmids, each construct was linearized with NotI and in vitro transcription was performed with the mMessage mMachine SP6 Transcription kit (Thermo Fisher Scientific) according to manufacturer's instructions. 150 pg *SLF2* mRNA with 6 ng *slf2* MO and 150 pg *SMC5* mRNA with 9 ng *smc5* MO was used for in vivo complementation assays.

**Fig. 7 | SLF2 and SMC5 patient cells exhibit mosaic variegated hyperploidy, mitotic abnormalities and sister chromatid cohesion defects. a** Quantification of the numbers of chromosomes per metaphase in peripheral blood lymphocytes from SLF2 or SMC5 patients, or an unrelated WT individual. 200 metaphases were counted in total from 2 independent blood samples. **b** Average number of mitotic cells with mis-segregated lagging chromosomes in SLF2 and SMC5 mutant fibroblast cell lines infected with lentiviruses encoding WT SLF2, WT SMC5, or an empty vector. $n = 3$ independent experiments for SLF2-P1, SMC5-P7 and SMC5-P8, and $n = 4$ independent experiments for SLF2-P2. A minimum of 250 mitotic cells were counted. **c** Representative images of mitotic cells from (**b**) with lagging chromosomes (scale bar: 10 μM). **d** Average number of mitotic cells with mis-segregated lagging chromosomes in U-2 OS SLF2 CRISPR HM cells infected with lentiviruses encoding WT SLF2 or an empty vector. $n = 3$ independent experiments. A minimum of 190 mitotic cells were counted. **e** Left: percentage of metaphases with rail-road chromosomes in peripheral blood lymphocytes from SLF2 or SMC5 patients, or an

unrelated WT individual. A minimum of 380 metaphases were counted in total from 2 independent blood samples. Right: Representative images of metaphases (scale bar: 10 μm). **f** Percentage of metaphases with premature chromatid separation following 4 h treatment with 25 μM MG132 in SLF2 and SMC5 patient LCLs. $n = 4$ independent experiments. 200 total metaphases were counted. **g** Percentage of S/G2 cells (CENPF positive cells) with >2 centrosomes with or without 24 h exposure to 250 nM APH. $n = 3$ independent experiments. A minimum of 900 CENPF positive cells were counted. **h** Percentage of mitotic cells in SLF2 and SMC5 mutant LCLs with multi-polar spindles in untreated cells and cells exposed to 250 nM APH for 24 h. A minimum of 300 mitotic cells were counted over 3 independent experiments. **i** The percentage of G1-phase cells (CENPF negative cells) with >5 53BP1 bodies in SLF2 and SMC5 mutant fibroblast cell lines, with or without 24 h exposure to 500 nM APH. $n = 4$ independent experiments. A minimum of 390 G1-phase cells were counted. In all cases, a Student's t test (two-sided, equal variance) statistical test was performed and error bars denote SEM.

## Live imaging of zebrafish larvae
Images of tricaine-anesthetized larvae at 3 dpf were captured using the Vertebrate Automated Screening Technology (VAST) Bioimager (Union Biometrica) mounted to an AXIO Imager.M2m microscope (Zeiss) with a *10x* objective lens. Larvae were passed sequentially through a 600 μm capillary on the detection platform. Each larva was detected by software on the computer screen and oriented automatically for lateral and ventral side images with a pre-provided template setting in the software. VAST software (version 1.2.6.7) operated in automatic imaging mode with a 70% minimum similarity threshold, as described[73]. Bright field lateral images were captured with the VAST onboard camera and a fluorescent signal from ventrally positioned larvae with an Axiocam 503 monochrome camera (Zeiss) and ZenPro software (Zeiss).

## TUNEL assay and phospho-histone H3 (pHH3) immunostaining in zebrafish larvae
Terminal deoxynucleotidyl transferase biotin-dUTP nick end labeling (TUNEL) assays or pHH3 immunostaining on whole-mount embryos were performed as described[27,74,75]. Embryos were dechorionated at 2 dpf (*slf2* and *smc5*) or 3 dpf (*smc5*) and fixed overnight in 4% paraformaldehyde (PFA) at 4 °C. Embryos were then dehydrated in methanol at −20 °C for 2 h and gradually rehydrated in methanol in PBS and 0.1% Tween (PBST) in the following percent volume/volume ratios: 75/25; 50/50; 25/75 for 10 min each at RT. Embryos were bleached for 12 min in a solution of 9 ml PBST + 1 ml H$_2$O$_2$ + 0.05 g KOH before proteinase K treatment and fixation in 4% PFA for 20 min at RT. For TUNEL, embryos were then incubated in equilibration buffer for 1 h and treated overnight with TdT enzyme at 37 °C in a humidified incubator. Following treatment with digoxigenin (ApopTag red in situ apoptosis detection kit, Sigma-Aldrich) for 2 h, embryos were washed *3x* with PBST (10 min each) and processed for imaging. For pHH3 staining, embryos were washed *3x* (10 min each) with PBST and incubated in blocking solution (IF buffer [1% BSA in PBST] + 10% FBS) for 1 h. Embryos were then treated with primary antibody diluted in 1% BSA overnight: anti-pHH3 (SCBT, sc-374669: 1:500) at 4 °C. Following staining with a secondary antibody: Alexa Fluor 488 goat anti-rabbit IgG (Thermo Fisher Scientific, A11008: 1:500) diluted in 1% BSA for 2 h at RT, embryos were washed *2x* (10 min each) with IF buffer and processed for imaging. For both TUNEL and pHH3 stained embryos, a z-stacked fluorescent signal of the dorsal aspect was captured with a Nikon AZ100 microscope facilitated by a Nikon camera controlled by Nikon NIS Elements Software.

## Zebrafish image analysis
ImageJ (NIH) was used to measure lateral head size, ceratohyal angle and count cells (TUNEL or pHH3) in the specified head region. Raw images were exported as TIF files and contrast and brightness were adjusted using identical settings for all images across the experiments.

To measure head size, a straight line was drawn from the posterior otolith to the tip of the mouth (line a), the dorsal head area outlined (line b), and the arbitrary shape closed with a line perpendicular to line a (line c). Ceratohyal angle was measured with the angle tool. To count TUNEL or pHH3 positive cells, the image-based tool for counting nuclei (ICTN) plugin for ImageJ was used. A consistent region between the two eyes was selected that spanned the most anterior region of the head to the most anterior region of the yolk.

## Immunofluorescence in human cells
Patient-derived fibroblasts or U-2 OS CRISPR HM cells were seeded onto coverslips at least 48 h before extraction and fixation. Cells were pre-extracted for 5 min on ice with ice-cold extraction buffer (25 mM HEPES [pH 7.4], 50 mM NaCl, 1 mM EDTA, 3 mM MgCl$_2$, 300 mM sucrose, and 0.5% Triton X-100) and then fixed with 4% paraformaldehyde (PFA) for 10 min. For immunofluorescence involving patient-derived LCLs, cells were seeded onto Poly-L-Lysine coated coverslips 20 min before fixation with ice-cold methanol for 20 min. For immunofluorescence using cells treated with exogenous DNA damage, patient-derived fibroblasts or LCLs cells were incubated with 500 nM APH, 50 ng/ml MMC or 250 μM CX5461 (Selleck Chemicals, S2684), as indicated in the figure legends, 24 h before fixation.

Fixed cells were then stained with primary antibodies specific to γH2AX (Sigma-Aldrich, 05-636; 1:1,000), CENPA (Abcam, Ab13939; 1:750), 53BP1 (Novus Biologicals, NB100−304; 1:1,000), CENPF/Mitosin (Abcam, Ab5; 1:500 and BD Transduction Laboratories, 610768; 1:500), α-Tubulin (Sigma-Aldrich, B-5-1−2; 1:4000), PCNT (Abcam, Ab4448; 1:100), and RAD51 (Merck, PC130; 1:500), and with secondary antibodies: anti-rabbit IgG Alexa Fluor 488 (Thermo Fisher Scientific, A11070; 1:1000) and anti-mouse IgG Alexa Fluor 594 (Thermo Fisher Scientific, A11032; 1:1000). Cells were then stained with DAPI and visualized with a *100x* oil-immersion objective lens on a Nikon Eclipse Ni microscope.

To visualize DNA replication, cells were incubated in medium containing 10 μM EdU for 30−45 min before harvesting. EdU immunolabeling was performed using the Click-iT EdU Imaging Kit (Thermo Fisher Scientific, C10337) according to the manufacturer's protocol.

## DNA fiber spreading assay
Patient-derived fibroblasts or U-2 OS cells were seeded at least 48 h prior to harvesting. Cells were incubated with 25 mM CldU for 30 min, washed with media containing 250 mM IdU (with or without 250 μM CX5461, 1 μM pyridostatin, 50 nM etoposide or 1 μM BMH21), incubated with 250 mM IdU (with or without 250 μM CX5461, 1 μM pyridostatin, 50 nM etoposide or 1 μM BMH21) for 30 min, and harvested by trypsinization. For patient-derived LCLs, untreated cells were incubated with 25 mM CldU for 20 min, washed with media containing 250 mM IdU, before being incubated with 250 mM IdU for 20 min and harvested. LCLs were incubated with 50 ng/ml MMC for 24 h prior to pulse

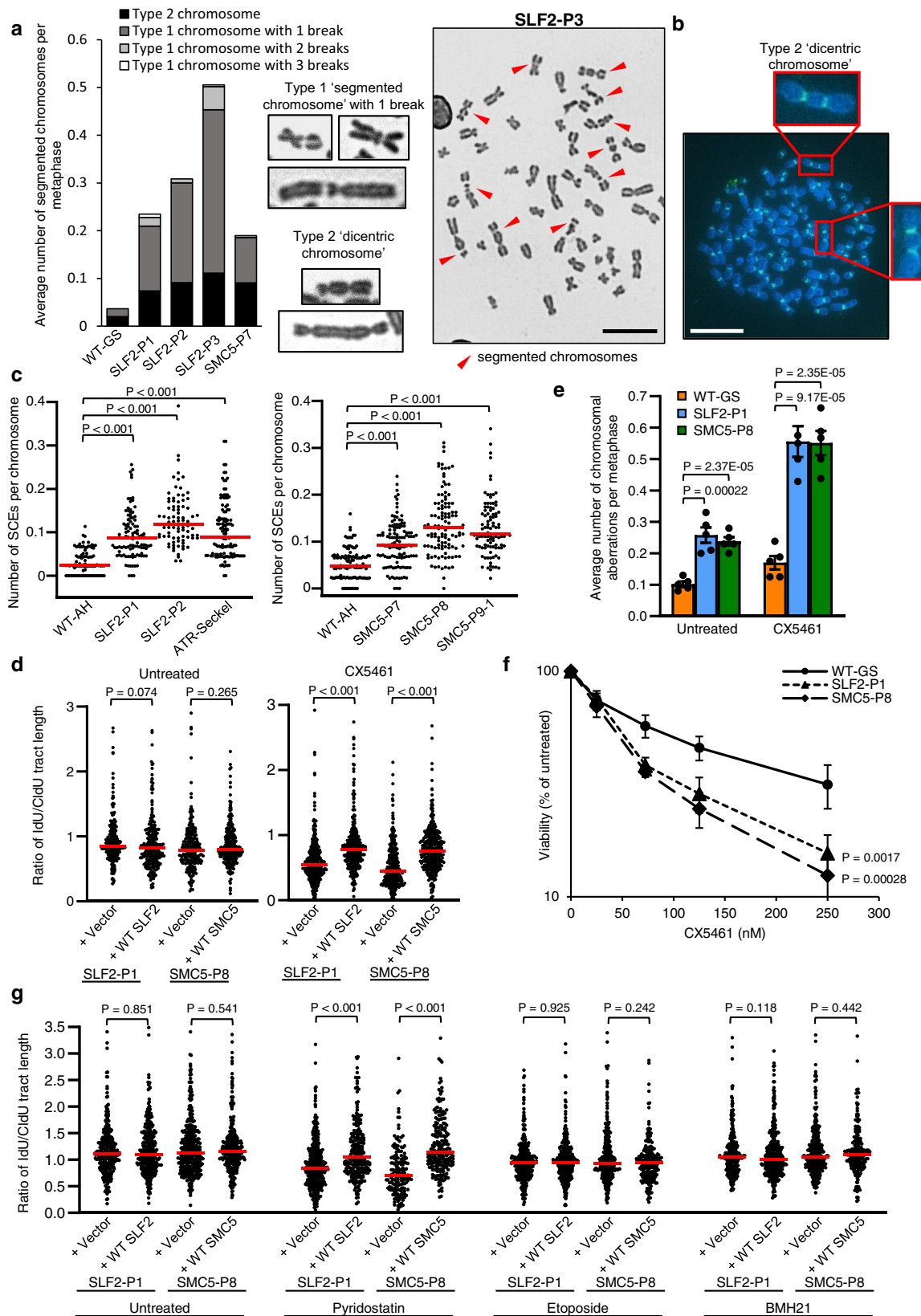

labeling with 25 mM CldU for 20 min and then 250 mM IdU for 20 min. For all incubation or washing steps, 50 ng/ml MMC was present in the media. For cells treated with HU, after being incubated with 25 mM CldU for 20 min, LCLs were incubated with media containing 2 mM HU for 2 h, before being washed in media containing 250 mM IdU, then incubated with 250 mM IdU for 20 min and harvested.

Following harvesting, cells were washed with PBS and resuspended to a concentration of 500,000 cells/ml in PBS, and then lysed in lysis buffer (200 mM Tris-HCl [pH 7.5], 50 mM EDTA, 0.5% SDS) directly on glass microscope slides. DNA fibers were spread down the slide by gravity, fixed in methanol/acetic acid (3:1) and denatured with 2.5 M HCl. The thymidine analogs, CldU and IdU, were detected via rat

**Fig. 8 | Variants in the RAD18-SLF1/2-SMC5/6 complex compromise the ability of cells to replicate in the presence of stabilized G4 quadruplex structures.** **a** Left: Average number of segmented chromosomes per metaphase in peripheral blood lymphocytes (PBLs) from SLF2 or SMC5 patients, or an unrelated WT individual. 250 total metaphases were counted from 2 independent blood samples. Middle: Representative images of 'type 1' and 'type 2' segmented chromosomes. Right: Representative image of a metaphase exhibiting segmented chromosomes from SLF2-P3 PBLs (scale bar: 10 μM). **b** Representative image of FISH with a centromere-specific probe showing dicentric chromosomes in a metaphase prepared from SLF2-P3 PBLs (scale bar: 10 μM). **c** Average number of sister chromatid exchanges in metaphase spreads from SLF2 and SMC5 patient-derived LCLs. $n = 3$ independent experiments. A minimum of 100 metaphases were counted. **d** Quantification of the IdU:CldU track length ratio in untreated and CX451-treated SLF2 and SMC5 patient fibroblast cells. Cell lines were pulse-labeled first with CldU for 30 min, followed by IdU, with or without 250 nM CX5461, for 30 min. $n = 3$ independent experiments. A minimum of 250 ongoing fork structures were counted. **e** Average number of chromosomal aberrations (chromatid/chromosome gaps, breaks, fragments and chromosome radials) per metaphase in SLF2 and SMC5 patient-derived LCLs with and without 24 h exposure to 250 nM CX5461. $n = 5$ independent experiments. A minimum of 350 metaphases were counted. Student's t test (two-sided, equal variance) was performed. Error bars denote SEM. **f** LCL proliferation assay. WT and SLF2 and SMC5 patient-derived LCLs were cultured in increasing concentrations of CX5461 for the time untreated cells took to undergo three population doublings. Cell viability following CX5461 treatment was calculated as a percentage of the number of untreated cells. $n = 4$ independent experiments. Error bars denote SEM. A two-way ANOVA statistical test was performed. **g** Quantification of IdU:CldU track length ratio in untreated, pyridostatin-, etoposide- and BMH21-treated SLF2 and SMC5 mutant fibroblast cells. Cell lines were pulse-labeled first with CldU for 30 min, followed by IdU with or without 1 μM pyridostatin, 50 nM etoposide or 1 μM BMH21, for 30 min. $n = 3$ independent experiments. A minimum of 150 ongoing forks were counted. For (**c**, **d**, **g**), red lines denote median values, and a Mann-Whitney rank sum statistical test was performed.

anti-BrdU antibody (clone BU1/75, ICR1; Abcam, ab6326; 1:500) and mouse anti-BrdU antibody (clone B44; BD Biosciences, 347583; 1:500) respectively, and secondary antibodies conjugated to Alexa Fluor 594 or Alexa Fluor 488 (Thermo Fisher Scientific). Labeled DNA fibers were visualized with a Nikon Eclipse Ni microscope with *100x* oil-immersion objective lenses, and images were acquired with NIS Elements software (Nikon Instruments). Replication fork structures and CldU and IdU track lengths were then quantified with ImageJ software (US NIH).

### Metaphase spreads

Giemsa-stained metaphase spreads from patient-derived cell lines or U-2 OS CRISPR SLF2 HM cells were prepared by adding of 0.2 mg/ml colcemid (KaryoMAX, Life Technologies) and incubating for 3 h. The cells were then harvested by trypsinization, subjected to hypotonic shock for 30 min at 37 °C in hypotonic buffer (10 mM KCl, 15% FCS), and fixed in ethanol/acetic-acid solution (3:1). The cells were dropped onto microscope slides, stained for 15 min in Giemsa-modified solution (Sigma-Aldrich; 5% vol/vol in water), and washed in water for 5 min. For analysis of cohesion fatigue in SLF2 patient LCLs, the metaphase spread protocol was followed as above. However, instead of adding colcemid, 25 μM MG132 (Sigma-Aldrich, M7449) was added 4 h before harvesting.

To prepare Giemsa-stained metaphase spreads from peripheral blood, whole blood was diluted in RPMI-1640 and 180 μg/ml PHA (Thermo Fisher Scientific) was added for 48–72 h at 37 °C. 4 h prior to harvesting 0.2 mg/ml colcemid was added. The cells were pelleted and then subjected to hypotonic shock for 10 min at 37 °C in hypotonic buffer (0.075M KCl). Finally, the cells were then fixed in methanol/acetic-acid solution (3:1) and processed as described above.

### Fluorescence in situ hybridization

For Fluorescence In Situ Hybridization (FISH) was carried out on peripheral blood lymphocytes metaphases using a peptide nucleic acid (PNA) pan-centromere FISH probe conjugated to Alexa Fluor 488 (5'-ATTCGTTGGAAACGGGA-3', PNA Bio, F3004 CENPB-Alexa488). Briefly, the PNA FISH probes was made up as per the manufacturer's instructions. Metaphase spreads were harvested from patient blood samples as above, and metaphases were dropped onto acetic-acid humidified microscope slides. 24 h later, the slides were rehydrated in PBS, dehydrated in an ethanol series (70%, 95%, 100%) and air dried. The slides were pre-warmed to 37 °C and before being incubated with hybridization buffer (20 mM Tris, pH7.4, 60% formamide, 0.5% blocking reagent [Roche Blocking Reagent, 11096176001], 1% v/v PNA probe) for 10 min at 85 °C. The slides were then incubated in a dark, humidified chamber at RT for 2 h, before being washed in wash buffer (70% formamide, 10-mM Tris) and dehydrated in an ethanol series (70%, 95%, 100%). The slides were then air dried and fixed with prolong gold DAPI mounting medium (ProLong Gold Antifade Mountant with DAPI, P36935).

### Sister chromatid exchange analysis

For sister chromatid exchange analysis, LCLs were incubated with 10 μM BrdU for 48 h before incubating with 0.2 μg/ml demecolcine for 3 h. Cells were then resuspended in 0.075M KCl and incubated at 37 °C for 1 h, fixed in methanol/acetic acid (3:1) and dropped onto microscope slides. The slides were then incubated in 10 μg/ml Hoescht for 20 min and exposed to UVA light for 1 h in 2× SSC buffer. Slides were incubated in 2× SSC buffer for 1 h at 60 °C and stained with 5% Giemsa. For metaphase spread analysis of cells treated with exogenous DNA damage, patient-derived LCLs cells were incubated with 500 nM APH or 50 ng/ml MMC 24 h before harvesting.

For analyses of telomere sister chromatid exchange, LCLs were cultured in the presence of BrdU:BrdC (final concentration of 7.5 mM BrdU (MP Biomedicals, 100166) and 2.5 mM BrdC (Sigma-Aldrich, B5002)) for 10 h prior to harvesting. KaryoMAX colcemid (Gibco, 15212-012) was added at a concentration of 0.1 μg/mL during the last 2 h. Cells were collected and washed in 75 mM KCl. Cells were then fixed *3x* in methanol:acetic acid (3:1) by adding fixative solution dropwise with constant gentle agitation by vortex. Following fixation, cells were dropped onto microscope slides and metaphase spreads were allowed to dry overnight. Next, slides were rehydrated in *1x* PBS and then treated with 0.5 mg/ml RNase A (Sigma-Aldrich, R5125) for 30 min at 37 °C. Next, slides were treated with 0.5 μg/ml Hoescht 33258 (Sigma-Aldrich, 861405) in *2x* SSC for 15 min at RT, UV-irradiated, and digested with ExoIII (NEB M0206L) for at least 30 min at 37 °C. Slides were then washed once in *1x* PBS and dehydrated in an ethanol series (70%, 90%, 100%) and air dried. FISH was performed using a TelC-Alexa488-conjugated PNA probe (PNA Bio, F1004; 1:1,000) followed by a TelG-Cy3-conjugated PNA probe (PNA Bio, F1006; 1:1,000) diluted in hybridization solution (10 mM Tris-HCl pH 7.2; 70% formamide; 0.5% blocking reagent (Roche, 11096176001)) each for 2 h at RT. Next, slides were washed at RT twice for 30 min in PNA wash A (70% formamide, 0.1% BSA, 10 mM Tris pH 7.2) and *3x* for 5 min in PNA wash B (100 mM Tris pH 7.2, 150 mM NaCl, 0.1% Tween-20). The second PNA wash B contained DAPI (Life Technologies, D1306) at a 1:1000 concentration. Slides were then dehydrated and dried as described above prior to mounting with Vectashield (Vectalabs, H1000). Slides were imaged using a Zeiss Spinning Disk confocal microscope. Image analyses were blinded and used FIJI version 2.1.0/153.c. Statistical analysis was performed using GraphPad Prism version 9.4.1.

### LCL proliferation assays

LCL proliferation assays were carried out as previously reported[49]. Briefly, LCLs were seeded at a concentration of $0.25 \times 10^6$ cells per ml in

25 cm² flasks and incubated with an increasing concentration of CX5461. The treated cells were counted when the untreated cells had reached a concentration of $2.0 \times 10^6$ cells per ml (approximately three population doubling times). The viability of the cells was expressed as a percentage of the untreated cell count.

## Plasmids, mutagenesis and sequencing primers

Total RNA was extracted from cell lines using RNeasy Mini kit (Qiagen) according to the manufacturer's instructions. DNA was removed by treatment with DNase I (Qiagen), and cDNA was generated using Superscript II and primed with oligo-dT (Thermo Fisher Scientific). PCR was carried out using Phusion Hot Start II (Thermo Fisher Scientific). *2x*Myc-SLF2 or untagged SMC5 lentiviral expression constructs were generated by cloning a PCR-generated cDNA into the NotI site of pLVX-IRES-neo (Takara Bio). The SLF2 and SMC5 ORFs were verified by sequencing using the primers in Supplementary Table 2.

Full length SLF2 cDNA was also cloned into pcDNA4/TO (Thermo Fisher Scientific) and deletion constructs were generated using KOD Hot Start DNA polymerase (Merck) according to manufacturer's instructions. The following primer sets in Supplementary Table 3 were used to generate the SLF2 deletion constructs and SLF2 'minimal binding region' (MBR) constructs. GFP-SLF2 is previously described[11]. Full length SMC5 cDNA was amplified and cloned into pEGFP-C1 (Takara Bio) using KpnI/BamHI. SLF2/SMC5 mutagenesis was achieved using the Q5 Site-Directed Mutagenesis Kit (E0554S, NEB) according to manufacturer's instructions. The following primer sets in Supplementary Table 3 were used to generate mutant expression vectors. SLF2 p.Gln1162His variant was generated using gene synthesis (Thermo Fisher Scientific).

Lentiviral plasmids encoding the bacterial Holliday junction resolvase RusA were a kind gift from Agata Smorgorzewska[40].

## RT-PCR analysis of patient cells

RT-PCR of SLF2 was performed using transcript specific primers (Supplementary Table 4) to assess the mRNA levels of the two longest annotated *SLF2* transcripts (NM_018121.4 and NM_001136123.2) in patient whole blood RNA (Paxgene) or commercially-obtained human cDNA panels: Human Universal QUICK-Clone II (Clontech), which is pool of cDNA obtained from 35 different healthy adult or fetal tissues; and Human multiple tissue cDNA (MTC) panel I (Clontech). PCR product was migrated on a 1% agarose gel for 40 min at 100 V.

## CRISPR-Cas9 genome editing of U-2 OS cells

Pairs of SLF2 targeting guide RNAs (sgRNA 1, 5'-AGTTTCAT-CACTCGGTTCCT-3'; sgRNA 2, 5'-GGCTTGGCACCTTCAAATTC-3') were designed using the CHOPCHOP web tool (version 2)[71,76] and hybridized and ligated into the purpose built AIO-GFP All-in-One Cas9D10A nickase vector at unique BbsI and BsaI sites. These constructs were transfected into U-2 OS cells using FuGENE transfection reagent according to manufacturer's instructions (3:1 ratio of FuGENE to DNA). Cells were sorted for high GFP expression by fluorescence-activated cell sorting (FACS) into 96-well dishes and recovered in McCoys 5A media supplemented with 20% FBS and 5% penicillin-streptomycin. After 3 weeks, 25 colonies were chosen to be propagated and screened for successful gene editing. After propagating, potential clones were lysed in lysis butter (100 mM Tris/HCl pH 8.5, 5 mM EDTA, 0.2% SDS, 200 mM NaCl, 100 μg Proteinase K/ml) and the DNA was precipitated with isopropanol and resuspended in 10 mM Tris/HCl, 0.1 mM EDTA, pH 7.5. Screening of genomic DNA from clones was achieved by sequencing a region of SLF2 surrounding the Cas9 nickase cut sites (Reverse primer, 5'-AGTTCCGATAATCCACCCCTT-3'; Forward primer, 5'-TTTCTGCAACCAGGTAGTCCT-3'). Following secondary screening of five clones by Western blotting, two SLF2 CRISPR HM clones were chosen (renamed as cl.1 and cl.2) and were

characterized further by inserting the amplified region of SLF2 described above into TOPO-TA vectors. 20 TOPO-TA vector clones were then sequenced for both cl.1 and cl.2 to identify all SLF2-mutant alleles and ensure no WT allele was present. The HM clones cl.1 and cl.2 were then complemented by 2xMyc-tagged SLF2 cloned into pLVX-IRES-neo (Takara Bio).

## Statistical analysis

Statistical analyses were performed as indicated in the figure legends. A *p* value of <0.05 indicates significance. The number of independent experimental replicates is denoted in the figure legends. In all cases, independent experiments represent distinct samples, and not the same sample measured repeatedly.

## Reporting summary

Further information on research design is available in the Nature Research Reporting Summary linked to this article.

## Data availability

The datasets generated during WES that support this study are available from the corresponding authors upon reasonable request. Informed consents from patients do not cover the deposition of sequencing data from the patient samples, but data can be shared for research purposes with permission of the patient or his/her legal guardian. Gene variant frequency was obtained from the gnomAD database (https://gnomad.broadinstitute.org/). Accession codes for genes/proteins analysed within this study are: Human SLF2 (NM_018121.4, NM_001136123.2, NP_060591.3), Human SMC5 (NM_015110.4, NP_055925.2), zebrafish slf2 (XM_002664123.6, XP_002664169.3), zebrafish smc5 (NM_001193541.1, NP_001180470.1). Plasmids obtained from Addgene (https://www.addgene.org/) used in this study: pLV-hTERT-IRES-hygro (Addgene #85140), psPax2 (Addgene #12260) and pMD2.G (Addgene #12259). PDB files used within this study to model the structural impact of SMC5 patient variants: Saccharomyces cerevisiae Smc5 3HTK [https://doi.org/10.2210/pdb3HTK/pdb], Pyrococcus furiosus RAD50 1F2T[https://doi.org/10.2210/pdb1F2T/pdb] and 1FTU [https://doi.org/10.2210/pdb1FTU/pdb]. AlphaFold models used to facilitate structural predictions: human SMC5 (AF-Q8IY18-F1). Source data are provided with this paper.

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

## Acknowledgements

We would like to thank the parents and affected individuals from the Atelís Syndrome families for taking part in this study and generously donating tissue samples. We would also like to thank Agata Smorgorzewska, Kasper Fugger and Stephen West for providing plasmids expressing RusA, Gen1, Mus81 and Eme1. G.S.S., R.H., G.S.M., S.L.C., A.G. are funded by a CR-UK Program Grant (C17183/A23303). L.J.G. is supported by a joint funded University of Birmingham and CR-UK Ph.D studentship (C17422/A25154). J.J.R. and A.M.R.T. are supported by the University of Birmingham. S.S.J. is supported by a project grant funded by the Great Ormond Street Hospital Charity and Sparks (V5019). R.F.S. and N.Mai are supported by Novo Nordisk Foundation (NNF14CC0001) and Independent Research Fund Denmark (9040-00038B). D.P. and A.P.J. are supported by a European Union Horizon 2020 research and innovation program European Research Council (ERC) Advanced Grant (grant agreement 788093) and by a Medical Research Council (MRC) Unit core grant (U127580972). N.Mat is supported by AMED (JP21ek0109486, JP21ek0109549, and JP21ek0109493). Y.U. is supported by JSPS KAKENHI (JP21K15907). A.K.B. and R.M.B. are supported by grants from the National Institutes of Health (R01 GM134681 and R35 GM141805). F.U. is funded by the Higher Education Commission of Pakistan under the International research support initiative program (IRSIP). E.E.D. is supported by US National Institutes of Health grant R01 MH106826 and is the Ann Marie and Francis Klocke MD Research Scholar.

## Author contributions

L.J.G., J.J.R., R.F.S., R.M.B., R.H., G.S.M., S.L.C., M.R.H., N.Mai, A-K.B., and G.S.S. designed and performed the cell biology experiments. F.U., X.L., T.K., and E.E.D. generated, performed, and supervised analysis on the *slf2* and *smc5* knockdown and knockout zebrafish. L.J.G., A.G., and S.S.J. designed and generated the *SLF2* CRISPR knockout constructs and U-2 OS cell lines. A.W.O carried out structural modeling of patient-associated mutations in SMC5. T.N. and N.Mat, designed and generated aid-tagged SLF2 degron cell lines. B.I., G.A.M-M., S.Ch, C.G.M., D.P., M.A.S., N.N., Z.Y., M.D., A.K., P.V., A.J., S.A.S., C.G-J., K.W.B., A.P.A.S., M.K., D.J., Y.U., Y.O., A.M., H.O., Z.A., J.A., C.T.R.M.S., A.M.R.T., A.P.J., and C.LeC. provided patient samples, performed next-generation sequencing, carried out bioinformatic analysis of next-generation sequencing data and performed other molecular genetic experiments. J.J.R., A.M.R.T., N.Mai, A-K.B., A.P.J., E.E.D. and G.S.S. wrote the paper. G.S.S. planned and supervised the study.

## Competing interests

The authors declare no competing interests.

## Additional information

Laura J. Grange [1,30], John J. Reynolds [1,30], Farid Ullah[2,3,30], Bertrand Isidor[4,30], Robert F. Shearer [5], Xenia Latypova[4],
Ryan M. Baxley [6], Antony W. Oliver [7], Anil Ganesh[1], Sophie L. Cooke [1], Satpal S. Jhujh [1], Gavin S. McNee[1],
Robert Hollingworth[1], Martin R. Higgs [1], Toyoaki Natsume[8], Tahir Khan [9], Gabriel Á. Martos-Moreno[10], Sharon Chupp[11],
Christopher G. Mathew [12], David Parry [13], Michael A. Simpson [14], Nahid Nahavandi[15], Zafer Yüksel [15],
Mojgan Drasdo[15], Anja Kron[15], Petra Vogt[15], Annemarie Jonasson[15], Saad Ahmed Seth [16], Claudia Gonzaga-Jauregui[17,18],
Karlla W. Brigatti[19], Alexander P. A. Stegmann [20], Masato Kanemaki [21], Dragana Josifova [22], Yuri Uchiyama [23,24],
Yukiko Oh[25], Akira Morimoto[25], Hitoshi Osaka [25], Zineb Ammous[11], Jesús Argente [10,26], Naomichi Matsumoto [23],
Constance T.R.M. Stumpel[27], Alexander M. R. Taylor [1], Andrew P. Jackson[13], Anja-Katrin Bielinsky [6], Niels Mailand [5],
Cedric Le Caignec [28] ✉, Erica E. Davis [2,29] ✉ & Grant S. Stewart [1] ✉

[1]Institute of Cancer and Genomic Sciences, University of Birmingham, Birmingham, UK. [2]Advanced Center for Genetic and Translational Medicine (ACT-GeM),
Stanley Manne Children's Research Institute, Ann & Robert H Lurie Children's Hospital of Chicago, Chicago, IL, USA. [3]National Institute for Biotechnology and
Genetic Engineering (NIBGE-C), Faisalabad, Pakistan Institute of Engineering and Applied Sciences (PIEAS), Islamabad, Pakistan. [4]Service de Génétique
Médicale, CHU Nantes, Nantes Cedex 1, France. [5]Novo Nordisk Foundation Center for Protein Research, Faculty of Health and Medical Sciences, University of
Copenhagen, Copenhagen, Denmark. [6]Department of Biochemistry, Molecular Biology and Biophysics, University of Minnesota, Minneapolis, MN, USA.
[7]Genome Damage and Stability Centre, Science Park Road, University of Sussex, Falmer, Brighton, UK. [8]Department of Chromosome Science, National
Institute of Genetics, Research Organization of Information and Systems (ROIS), Mishima, Shizuoka, Japan. [9]Center for Human Disease Modeling, Duke
University Medical Center, Durham, NC, USA. [10]Hospital Infantil Universitario Niño Jesús, CIBER de fisiopatología de la obesidad y nutrición (CIBEROBN),
Instituto de Salud Carlos III, Universidad Autónoma de Madrid, Madrid, Spain. [11]The Community Health Clinic, Topeka, IN, USA. [12]Sydney Brenner Institute for
Molecular Bioscience, Faculty of Health Sciences, University of the Witwatersrand, Johannesburg, South Africa. [13]MRC Human Genetics Unit, MRC Institute of
Genetics and Molecular Medicine, Western General Hospital, The University of Edinburgh, Edinburgh, Scotland. [14]Department of Medical and Molecular
Genetics, Faculty of Life Science and Medicine, Guy's Hospital, King's College London, London, UK. [15]Bioscientia Institute for Medical Diagnostics, Human
Genetics, Ingelheim, Germany. [16]King Fahad Military Medical Complex, Dhahran, Saudi Arabia. [17]Regeneron Genetics Center, Regeneron Pharmaceuticals
Inc., Tarrytown, NY, USA. [18]International Laboratory for Human Genome Research, Universidad Nacional Autónoma de México, Querétaro, México. [19]Clinic for
Special Children, Strasburg, PA, USA. [20]Department of Clinical Genetics, Maastricht University Medical Center, Maastricht, The Netherlands. [21]Department of
Genetics, The Graduate University for Advanced Studies (SOKENDAI), Mishima, Shizuoka, Japan. [22]Clinical Genetics Department, Guy's Hospital, London, UK.
[23]Department of Rare Disease Genomics, Yokohama City University Hospital, Yokohama, Japan. [24]Department of Human Genetics, Yokohama City University
Graduate School of Medicine, Yokohama, Japan. [25]Department of Paediatrics, Jichi Medical University School of Medicine, Tochigi, Japan. [26]IMDEA Ali-
mentación/IMDEA Food, Madrid, Spain. [27]Department of Clinical Genetics and GROW-School for Oncology and Developmental Biology, Maastricht University
Medical Center, Maastricht, The Netherlands. [28]Centre Hospitalier Universitaire Toulouse, Service de Génétique Médicale and ToNIC, Toulouse NeuroImaging
Center, Inserm, UPS, Université de Toulouse, Toulouse, France. [29]Department of Pediatrics; Department of Cell and Developmental Biology, Feinberg School
of Medicine, Northwestern University, Chicago, IL, USA. [30]These authors contributed equally: Laura J. Grange, John J. Reynolds, Farid Ullah, Bertrand Isidor.
✉e-mail: lecaignec.c@chu-toulouse.fr; eridavis@luriechildrens.org; g.s.stewart@bham.ac.uk

