## [Peer Review File · Nature Communications]

Pathogenic variants in SLF2 and SMC5 cause segmented chromosomes and mosaic variegated hyperploidy.REVIEWER COMMENTS

Reviewer #1 (Remarks to the Author):

This is an interesting and extensive analysis of the phenotypes associated with patients with mutations in SLF2 and SMC5. SMC5 is a core component of the essential and conserved structural maintenance of chromosomes complex SMC5/6 complex required to regulate homologous recombination and the responses to replication stress. It also has less well understood roles in regulating transcription and as a restriction factor for hepatitis B. SLF2 was identified as a recruiter of the SMC5/6 complex to sites of DNA damage and is functionally equivalent to Nse5 and Nse6.

In this study eleven patients were identified with microcephaly and short stature, nine with mutations in SLF2 and two with mutations in SMC5. The microcephaly phenotype was elegantly supported using zebra fish models showing that the microcephaly was due the SLF2 and SMC5 mutations. This is consistent with work from the Jordan lab (Atkins et al 2020) that showed that conditional SMC5 knockouts in mice exhibited neurodevelopmental defects, elevated replication stress, increased chromosome mis-segregation. Atkins et al also showed that neurodevelopmental defects could be suppressed by deletion of p53 or CHEK2, indicating that these phenotypes were due to increased apoptosis resulting from activation of the DNA damage checkpoint. Thus, while the zebra fish analysis is of a high quality, the results are not novel.

In this manuscript the authors went on analyse the cellular defects of the patients and in particular the DNA damage response in more detail. The mutations in SLF2 resulted in major deletions/disruption of SLF2 and loss of interaction with RAD18 and the SMC5/6 complex but the mutations in SMC5 led to minimal differences to the protein and to protein/protein interactions. However in both cases the SMC5/6 complex was no longer recruited to laser tracks indicating a lack of recruitment to DNA damage. The SLF2 cellular phenotype is consistent with SLF2 being required to recruit SMC5/6 to sites of DNA damage. However, it is not clear why the mutations in SMC5 would have the same effect and mechanistic insight is lacking.

Patient-derived cell lines and U2-OS SLF2 cell lines showed increased replication stress by DNA fibre analysis and, similarly to SMC5/6 shut off cell lines, mitotic abnormalities (Bueno Venegas et al, 2020 should be referenced here) which were further characterized in detail. The chromosome segregation defects, cohesin mis-regulation and lack of recruitment of Mus81 and Yen1 were first characterized in detail by Copsey et al, 2013, and this should also be referenced.

The analysis of G4 structures is not particularly convincing. A slight increase (0.25-0.55) in the number of aberrant chromosomes is seen after addition of a G4 intercalating drug but only a very small proportion of this is due to segmented chromosomes (0.02-0.08) which the authors correlate with the presence of unresolved recombination intermediates and in addition no survival curves are presented to show the consequences on cell viability.

In summary this is an elegant and detailed analysis of the phenotypes of patients with a new syndrome due to mutations in SLF2 or SMC5. The novelty is in the identification of the patients. The clinical and cellular phenotypes are consistent with previous analyses in mice, human and yeast. The SLF2 phenotypes due to mutations which lead to a major disruption of the protein are consistent with SLF2 being required to recruit SMC5/6 to sites of DNA damage but mechanistic insight into the SMC5 phenotypes is lacking. It is not clear why when the mutations do not disrupt interactions with SLF2 the SMC5/6 complex is not recruited to DNA damage.

Reviewer #2 (Remarks to the Author):

The authors identify a new genetic disorder caused by mutations in SLF2 or SMC5 that results in a rather unique phenotype, clinically overlapping with Fanconi Anemia (FA), the Warsaw Breakage Syndrome (WABS) cohesinopathy, and the mosaic variegated aneuploidy (MVA) syndrome. The work is very comprehensive, covering genetic identification of mutations in patients, establishment of cell lines and zebrafish models of SLF2 and SMC5 mutations, their complementation, biochemical assays of protein interactions in the context of the described RAD18-SLF1/2-SMC5/6 complex, analysis of replication defects, mitotic abnormalities, chromosome aberrations in the presence or absence of additional replication stress or DNA damage, with or without complementation. It is overall an impressive piece of work, which deserves publication in Nature Communications. However, the choice of references in the SMC5/6 field is often quite poor, and critical papers that should be discussed are often not even mentioned. This can be easily corrected and I will give suggestions in this regard along with some comments I have for several panels and for an additional experiment.

1) Lines 170-188. Here the phenotype of patients is being narrated, with the observation that patients develop an atypical form of FA. This is interesting and should be discussed in the context of a recent paper, PMID: 31867888, in which SMC5/6 physical and functional interaction with FANCD2-I is being described. This reference also shows that SMC5/6 functions jointly with DDX11 in the repair of DNA lesions, a finding that will become of interest later.

2) Lines 170-188. A phenotype of atrial and ventricular defects is not common in patients with DNA replication disorders, but common in cohesinopathies (see PMID: 31516082; PMID: 26420840). SMC5/6 has tight interconnections with cohesin, as I would detail at several points below.

3) Lines 230-239. Analysis of SMC5 mutations reveals that their main defect is not in failure to assemble RAD18-SLF1/2-SMC5 complexes, but rather to be efficiently retained to sites of DNA damage. Recently such mutants have been described in budding yeast, both during unperturbed conditions at NPSs and upon replication stress, and the analysis revealed that such hypomorph mutants are relevant for SMC5/6 biology (PMID: 33833229). This reference will become very important later on, during the analysis of the chromosome abnormalities and discovery of the segmented chromosome phenotype.

4) Figures 5 and Supplementary 5, lines 291-299. The description states that there are no gross alterations in forks speed. However, in Supplementary Fig 5b, it seems that with the exception of SMC5-P7, both P8 and P9 samples show a strong decrease in fork speed. For Fig. 5c-d, an explanation on how fork asymmetry was measured is needed, along with one example. Moreover, the same panel of stalled fork example is used 5 times in Fig. 5 and twice in Supplementary Fig. 5. Please show other examples, or just show this only once in Fig. 5a.

5) Lines 313-315. The reported results on normal ATR, CHK1 activation and FANCD2-I ubiquitylation are in line with those shown recently in PMID: 31867888 (see point 1) and should be cited.

6) Figure 6. A typical image of the phenotype plotted needs to be shown for each of the main 4 phenotypes being characterized in panels a-e. Panels f-i of Fig. 6 describes chromosome aberrations, which are perhaps chromosome breaks. A change in the label will be needed, making the results in Figure 7 of even more interest.

7) Lines 319-320. Increased micronuclei are observed, in line with results reported in two recent studies (PMID: 31867888, 32320646).

8) Lines 345-346. Increased lagging chromosomes have been reported upon SMC5/6 depletion in 2 recent studies (PMID: 31867888, 32320646).

9) Lines 352-360. Several reports document a role for SMC5/6 in cohesion PMID: 19502785, PMID: 21245390, PMID: 27798241). The role of SMC5/6 in centromeric cohesion documented in the latest reference can provide an explanation for the rail-road phenotype and the intrinsic defect in fully activating the spindle-assembly checkpoint, explaining the PCS phenotype observed after treatment with MG132.

10) Page 390-392. The function of Smc5/6 in resolving recombination intermediates has been importantly documented and characterized in budding yeast, both after replication stress and under normal replication conditions (PMID: 17081974, 26698660, 15793567, 33833229) in papers that led the field and need to be cited. If the chromosome abnormalities are due to deficiency in resolving

recombination intermediates, the phenotypes could be alleviated by expression of bacterial RusA or GEN1, which could be of high interest.

11) Page 400. A role for SMC5/6 at specific subsets of DNA lesions and genomic sites prone to forming secondary structures has been described before (PMID: 26698660) and needs to be cited.

12) Page 405-406. DDX11 was shown to function with SMC5/6 in DNA repair (PMID: 31867888), and facilitate HR non-redundantly with BRCA1/2 by its action at a subset of genomic regions that likely contain G4 structures (PMID: 33879618, 32705708).

13) Line 416. These data suggest (rather than identify) as all G4 stabilizers are likely to have other effects not fully covered in the study.

14) Line 479. A role for Smc5/6 at fork pause sites and the rDNA locus requires citation of PMID: 26698660, 15793567.

15) Lines 483-485. A role for SMC5/6 in orchestrating the functions of both dissolvases and resolvases has been proposed in budding yeast (PMID: 33833229). This could indeed explain the type I segmented chromosomes, which may be rescued by expression of bacterial RusA or GEN1, as suggested at point 10 above.

Reviewer #3 (Remarks to the Author):

The manuscript Pathogenic variants in SLF2 and SMC5 cause segmented chromosomes and mosaic variegated hyperploidy is well written and a great amount of work to describe two new disease genes. I am in favor of publication with some minor modifications. These mainly stem around the conclusion that SLF2 and SMC5 are the causative variants for each of these patients – there is not sufficient information for the reviewers to assess this. Specific comments are below.

- Please include tables in supplementary for each patient that describe which variants were ruled out and why to convince the reviewer the due diligence has been done for each patient.

- Figure 1: The table should have added control allele frequencies from gnomAD for each variant, as well as an in silico score or two

- The way the first patient is separated out from the others doesn't make sense – this is a cohort describe it as a cohort.

- No note of segregation in P1, was this done. P5 is missing segregation given not available, was CNV analysis done to confirm there is no deletion in trans given its homozygous?

- Protein nomenclature should include brackets as per HGVS. This is missing in many figures and some places in the text. Please correct.

- Figure 2. All of these Western blots are over-exposed and not possible to draw conclusions on. Do the authors have better exposures from which to draw conclusions? Also a loading control not in this pathway should be shown or total protein staining.

Reviewer #1 (Remarks to the Author):

This is an interesting and extensive analysis of the phenotypes associated with patients with mutations in SLF2 and SMC5. SMC5 is a core component of the essential and conserved structural maintenance of chromosomes complex SMC5/6 complex required to regulate homologous recombination and the responses to replication stress. It also has less well understood roles in regulating transcription and as a restriction factor for hepatitis B. SLF2 was identified as a recruiter of the SMC5/6 complex to sites of DNA damage and is functionally equivalent to Nse5 and Nse6.

In this study eleven patients were identified with microcephaly and short stature, seven with mutations in SLF2 and four with mutations in SMC5. The microcephaly phenotype was elegantly supported using zebra fish models showing that the microcephaly was due the SLF2 and SMC5 mutations. This is consistent with work from the Jordan lab (Atkins et al 2020) that showed that conditional SMC5 knockouts in mice exhibited neurodevelopmental defects, elevated replication stress, increased chromosome mis-segregation. Atkins et al also showed that neurodevelopmental defects could be suppressed by deletion of p53 or CHEK2, indicating that these phenotypes were due to increased apoptosis resulting from activation of the DNA damage checkpoint. Thus, while the zebra fish analysis is of a high quality, the results are not novel.

Response: We thank the reviewer for their positive comments regarding the quality of our data. However, with respect to the reviewer's comment questioning the novelty of our data generated using zebrafish, we would like to point out that the SLF2/SMC5 knockout/knock down animal models were primarily generated to confirm the pathogenicity of the gene mutations identified in our patient cohort rather than to create novel knockout/knockdown animal models of the SMC5/6 pathway. Since neither loss of *SLF2* nor *SMC5* have been previously modelled in zebrafish, it was necessary for us to characterise the phenotypic abnormalities exhibited by these mutant animals and to examine possible cellular mechanisms underlying the development of the phenotypes. The fact that the developmental and cellular phenotype of the SMC5 knockdown zebrafish is consistent with what has been previously observed by the Jordan lab using conditional knockout mice, strongly supports the use of zebrafish to model patient-associated mutations in *SMC5*. However, loss of SLF2 has not been modelled in mice and as such, the generation of an SLF2 knockout animal is novel. Furthermore, we believe that our demonstration that loss of SLF2 recapitulates the same developmental deficits as loss of SMC5 in an animal model, highlights the functional importance of this component of the SMC5/6 pathway, for which little is known.

Therefore, we would suggest that the novelty does not lie with generating the animal models *per se*, but rather lies with the demonstration that the loss of SLF2, or expression of the patient-associated missense mutations in *SMC5*, are associated with the development of microcephaly in an animal model, and that this is caused by a G2/M arrest and an increase in apoptosis in the developing brain.

In this manuscript the authors went on to analyse the cellular defects of the patients and in particular the DNA damage response in more detail. The mutations in SLF2 resulted in major deletions/disruption of SLF2 and loss of interaction with RAD18 and the SMC5/6 complex but the mutations in SMC5 led to minimal differences to the

protein and to protein/protein interactions. However in both cases the SMC5/6 complex was no longer recruited to laser tracks indicating a lack of recruitment to DNA damage. The SLF2 cellular phenotype is consistent with SLF2 being required to recruit SMC5/6 to sites of DNA damage. However, it is not clear why the mutations in SMC5 would have the same effect and mechanistic insight is lacking.

Response: To address possible underlying reasons as to why the identified mutations in SMC5 compromise its recruitment/retention at sites of DNA damage, we initially carried out co-immunoprecipitation studies to assess whether the DelR372 or H990D SMC5 mutations affect binding to other subunits within the SMC5/6 complex. As a result of this analysis, we have demonstrated that loss of R372 destabilises the association of NSMCE2 with the rest of the SMC5/6 complex (Figure 2e and 2f). This indicates that the inability of the DelR372 SMC5 mutant to be recruited to sites of damage may be caused by a reduction or loss of its SUMO ligase activity comparable to what has been previously observed in NSMCE2 mutant patients/mice (Payne et al. 2014 JCI 124:4028-4038; Jacome et al. 2015 EMBOJ 34:2604-2619).

Unlike DelR372, the H990D mutation did not affect the binding of NSMCE2 to the SMC5/6 complex. Therefore, to provide a possible explanation for the inability of the H990D SMC5 mutant to be relocalised to sites of DNA damage, we used structural modelling to assess whether this mutation might affect the structure of the SMC5/6 complex. Interestingly, this analysis suggests that the H990D mutation may affect the ability of SMC5 to bind and/or turn over ATP. Whilst we have not formally tested this prediction, as this would require a large amount of additional work to purify the mutant SMC5/6 complex and assay its ATPase activity, which we believe is beyond the scope of this manuscript, it does offer some possible mechanistic insight as to why this mutation affects recruitment/retention of this complex at sites of DNA damage.

Patient-derived cell lines and U2-OS SLF2 cell lines showed increased replication stress by DNA fibre analysis and, similarly to SMC5/6 shut off cell lines, mitotic abnormalities (Bueno Venegas et al, 2020 should be referenced here) which were further characterized in detail. The chromosome segregation defects, cohesin mis-regulation and lack of recruitment of Mus81 and Yen1 were first characterized in detail by Copsey et al, 2013, and this should also be referenced.

Response: We thank the reviewer for highlighting these references and have added them to the text.

The analysis of G4 structures is not particularly convincing. A slight increase (0.25-0.55) in the number of aberrant chromosomes is seen after addition of a G4 intercalating drug but only a very small proportion of this is due to segmented chromosomes (0.02-0.08) which the authors correlate with the presence of unresolved recombination intermediates and in addition no survival curves are presented to show the consequences on cell viability.

Response: Whilst we appreciate the reviewer's concerns regarding the level of chromosome breakage induced by CX5461 in the SLF2/SMC5 mutant cells, we would like to point out that an increase of 0.25 to 0.55 chromosome aberrations per metaphase equates to a 2-fold increase, which we believe represents a significant increase in chromosomal aberrations.

Despite this, we agree with the reviewer that the level of segmented chromosomes induced by CX5461 is low. However, we would like to point out that the segmented chromosomes are predominantly observed in primary T cells derived from blood samples. We believe that the discrepancy between the high level of segmented chromosomes observed in the primary T cells compared to those observed in other immortalised cell lines (such as LCLs) may be due to the fact that circulating T cells are predominantly quiescent and only cycle when we stimulate them with PHA. Following a short level of stimulation, the T cells undergo 1-2 rounds of synchronous cell division before being blocked in mitosis with colcemid. This short round of cycling may reduce the ability of S/G2-phase-specific recombination pathways to resolve these recombination intermediates. However, in contrast, the immortalised LCLs are continually cycling and as such, this may increase the ability of back up anti-recombination pathways to deal with the segmented chromosomes. Based on this, it is difficult to determine the overall contribution of the increased level of segmented chromosomes to the genome instability present in patient-derived cell lines. Therefore, to address the reviewer's concerns, we have changed the text to tone down the proposed link between the unresolved G4 structures and the appearance of the segmented chromosomes

However, to strengthen the links between the SMC5/6 complex and G4 structures, we have carried out DNA fibre analysis and chromosome breakage with pyridostatin, another well characterised G4-quadruplex stabilising agent. This analysis demonstrates that pyridostatin also induces increased replication stress and genome instability in cells with mutations in SLF2 and SMC5 (Figure 8g, Supplementary Figure 20d). Additionally, we have shown that LCLs with mutations in SLF2 and SMC5 are sensitive to CX5461 treatment (Figure 8f). Furthermore, since CX5461, a known RNA Pol I inhibitor, has more recently been shown also to poison TOP2 (Pan et al. 2021. Nature Communications, 12:6468), we have also carried out DNA fibre analysis and chromosome breakage using well characterised inhibitors of RNA pol I (BMH21) and TOP2 (etoposide). Interestingly, this analysis demonstrated that SLF2 and SMC5 patient cells can replicate normally following RNA pol I or TOP2 inhibition (Figure 8g, Supplementary Figure 20d). Taken together, we believe that this provides strong evidence that the SMC5/6 complex functions to specifically resolve G4 structures encountered during DNA replication.

In summary this is an elegant and detailed analysis of the phenotypes of patients with a new syndrome due to mutations in SLF2 or SMC5. The novelty is in the identification of the patients. The clinical and cellular phenotypes are consistent with previous analyses in mice, human and yeast. The SLF2 phenotypes due to mutations which lead to a major disruption of the protein are consistent with SLF2 being required to recruit SMC5/6 to sites of DNA damage but mechanistic insight into the SMC5 phenotypes is lacking. It is not clear why when the mutations do not disrupt interactions with SLF2 the SMC5/6 complex is not recruited to DNA damage.

Response: We thank the reviewer for their comments. As outlined above, we believe that we have provided some mechanistic insight for why the DelR372 SMC5 mutation prevents recruitment/retention of the SMC5/6 complex at sites of DNA damage (loss of NSMCE2 binding) and a potential explanation for why this is also the case for the H990D SMC5 mutation (disruption of ATP binding and/or turnover).

Reviewer #2 (Remarks to the Author):

The authors identify a new genetic disorder caused by mutations in SLF2 or SMC5 that results in a rather unique phenotype, clinically overlapping with Fanconi Anemia (FA), the Warsaw Breakage Syndrome (WABS) cohesinopathy, and the mosaic variegated aneuploidy (MVA) syndrome. The work is very comprehensive, covering genetic identification of mutations in patients, establishment of cell lines and zebrafish models of SLF2 and SMC5 mutations, their complementation, biochemical assays of protein interactions in the context of the described RAD18-SLF1/2-SMC5/6 complex, analysis of replication defects, mitotic abnormalities, chromosome aberrations in the presence or absence of additional replication stress or DNA damage, with or without complementation. It is overall an impressive piece of work, which deserves publication in Nature Communications. However, the choice of references in the SMC5/6 field is often quite poor, and critical papers that should be discussed are often not even mentioned. This can be easily corrected and I will give suggestions in this regard along with some comments I have for several panels and for an additional experiment.

Response: We thank the reviewer for their positive comments regarding our paper. We are more than happy to include additional key references suggested by the reviewer.

1) Lines 170-188. Here the phenotype of patients is being narrated, with the observation that patients develop an atypical form of FA. This is interesting and should be discussed in the context of a recent paper, PMID: 31867888, in which SMC5/6 physical and functional interaction with FANCD2-I is being described. This reference also shows that SMC5/6 functions jointly with DDX11 in the repair of DNA lesions, a finding that will become of interest later.

Response: We thank the reviewer for highlighting this reference and have added it to the text.

2) Lines 170-188. A phenotype of atrial and ventricular defects is not common in patients with DNA replication disorders, but common in cohesinopathies (see PMID: 31516082; PMID: 26420840). SMC5/6 has tight interconnections with cohesin, as I would detail at several points below.

Response: We thank the reviewer for highlighting these references and have added them to the text.

3) Lines 230-239. Analysis of SMC5 mutations reveals that their main defect is not in failure to assemble RAD18-SLF1/2-SMC5 complexes, but rather to be efficiently retained to sites of DNA damage. Recently such mutants have been described in budding yeast, both during unperturbed conditions at NPSs and upon replication stress, and the analysis revealed that such hypomorph mutants are relevant for SMC5/6 biology (PMID: 33833229). This reference will become very important later on, during the analysis of the chromosome abnormalities and discovery of the segmented chromosome phenotype.

Response: We thank the reviewer for highlighting this reference and have inserted it within the text where we thought it was most relevant.

4) Figures 5 and Supplementary 5, lines 291-299. The description states that there are no gross alterations in fork speed. However, in Supplementary Fig 5b, it seems that with the exception of SMC5-P7, both P8 and P9 samples show a strong decrease in fork speed. For Fig. 5c-d, an explanation on how fork asymmetry was measured is needed, along with one example. Moreover, the same panel of stalled fork example is used 5 times in Fig. 5 and twice in Supplementary Fig. 5. Please show other examples, or just show this only once in Fig. 5a.

Response: We thank the reviewer for pointing out our misrepresentation of the fork speeds displayed by the SLF2/SMC5 mutant LCLs. We have amended the text to more accurately described the differences in replication fork speed exhibited by the SLF2 and SMC5 mutant LCLs.

Replication fork asymmetry represents the ratio of the left to right fork-track lengths of bidirectional replication forks. We had added a sentence to the figure legend stating this. An image of symmetrical and asymmetrical bi-directional forks have been added.

The image of the ongoing and stalled forks were merely added to demonstrate to the reader what type of replication fork structures was being quantified. The images were not meant to be indicative of specific replication forks structures from each set of experiments. We have therefore shown the replication structure images once and removed all duplicates of these images.

5) Lines 313-315. The reported results on normal ATR, CHK1 activation and FANCD2-I ubiquitylation are in line with those shown recently in PMID: 31867888 (see point 1) and should be cited.

Response: We have added the suggested reference to the text.

6) Figure 6. A typical image of the phenotype plotted needs to be shown for each of the main 4 phenotypes being characterized in panels a-e. Panels f-i of Fig. 6 describes chromosome aberrations, which are perhaps chromosome breaks. A change in the label will be needed, making the results in Figure 7 of even more interest.

Response: We have added representative images for each of the phenotypes quantified in Figure 6 to the supplementary data due to space limitations for figures (Supplementary Figure 14). In figure 6, the quantification of chromosomal aberrations per metaphase includes all types of chromosomal aberration i.e. chromatid/chromosome gaps, breaks, fragments and chromosome radials. This analysis does not include quantitative alterations in chromosome number or railroad chromosomes. These were quantified separately in Figure 7a, 7e, supplementary figures 12 and 13. We have added a note to the figure legend to indicate what has been included in the quantification, and have clarified this in the text. In figure 8a, only the segmented and dicentric chromosomes were quantified.

7) Lines 319-320. Increased micronuclei are observed, in line with results reported in two recent studies (PMID: 31867888, 32320646).

Response: We have added the suggested references to the text.

8) Lines 345-346. Increased lagging chromosomes have been reported upon SMC5/6 depletion in 2 recent studies (PMID: 31867888, 32320646).

Response: We have added the suggested references to the text.

9) Lines 352-360. Several reports document a role for SMC5/6 in cohesion (PMID: 19502785, PMID: 21245390, PMID: 27798241). The role of SMC5/6 in centromeric cohesion documented in the latest reference can provide an explanation for the railroad phenotype and the intrinsic defect in fully activating the spindle-assembly checkpoint, explaining the PCS phenotype observed after treatment with MG132.

Response: We thank the reviewer for highlighting these references and have added them to the text.

10) Lines 390-392. The function of Smc5/6 in resolving recombination intermediates has been importantly documented and characterized in budding yeast, both after replication stress and under normal replication conditions (PMID: 17081974, 26698660, 15793567, 33833229) in papers that led the field and need to be cited. If the chromosome abnormalities are due to deficiency in resolving recombination intermediates, the phenotypes could be alleviated by expression of bacterial RusA or GEN1, which could be of high interest.

Response: We thank the reviewer for highlighting these references and have added them to the text. To address whether the segmented chromosomes present in cells with mutations in the SMC5/6 complex result from a failure to resolve recombination intermediates, we obtained a RusA lentiviral expression plasmid from Prof. Agata Smogorzewska's lab (Garner et al. 2013 Cell Reports 5:207-215). Whilst we found the stable expression of WT RusA in patient-derived fibroblasts seemed to be quite toxic, we were eventually able to establish cell lines stably expressing low levels of WT RusA. Using these cells, we were able to show that expression of WT RusA in ATS cell lines increased the level of chromosomal aberrations in those complemented with an empty vector but not those complemented with either WT SLF2 or SMC5 (Supplementary Figure 19d). This would indicate that ATS cells exhibit higher levels of unresolved HR intermediates that are capable of being cleaved by the bacterial resolvase RusA. This would suggest that the failed resolution of HR intermediates represents a significant contributing factor to the development of this disease. However, the segmented chromosomes were present at very low levels in patient fibroblasts and proved to be too low to make any strong conclusions regarding the origins of this chromosome abnormality.

11) Line 400. A role for SMC5/6 at specific subsets of DNA lesions and genomic sites prone to forming secondary structures has been described before (PMID: 26698660) and needs to be cited.

Response: We have added the suggested reference to the text.

12) Lines 405-406. DDX11 was shown to function with SMC5/6 in DNA repair (PMID: 31867888), and facilitate HR non-redundantly with BRCA1/2 by its action at a subset of genomic regions that likely contain G4 structures (PMID: 33879618, 32705708).

Response: We thank the reviewer for highlighting these references and have added them to the text.

13) Line 416. These data suggest (rather than identify) as all G4 stabilizers are likely to have other effects not fully covered in the study.

Response: We have altered line 416 as suggested.

14) Line 479. A role for Smc5/6 at fork pause sites and the rDNA locus requires citation of PMID: 26698660, 15793567.

Response: We have added the suggested references.

15) Lines 483-485. A role for SMC5/6 in orchestrating the functions of both dissolvases and resolvases has been proposed in budding yeast (PMID: 33833229). This could indeed explain the type I segmented chromosomes, which may be rescued by expression of bacterial RusA or GEN1, as suggested at point 10 above.

Response: We have added the suggested reference.

Reviewer #3 (Remarks to the Author):

The manuscript Pathogenic variants in *SLF2* and *SMC5* cause segmented chromosomes and mosaic variegated hyperploidy is well written and a great amount of work to describe two new disease genes. I am in favor of publication with some minor modifications. These mainly stem around the conclusion that *SLF2* and *SMC5* are the causative variants for each of these patients – there is not sufficient information for the reviewers to assess this.

Specific comments are below.

- Please include tables in supplementary for each patient that describe which variants were ruled out and why to convince the reviewer the due diligence has been done for each patient.

- Figure 1: The table should have added control allele frequencies from gnomAD for each variant, as well as an in silico score or two.

Response: We thank the reviewer for their positive response to our manuscript. We have included a shortlist of all the potentially pathogenic gene variants identified by WES in *SLF2/SMC5* mutant patients (Supplementary Tables S2 - S10). These were filtered according to zygosity and in most cases, whether the variants segregated with parents. We have also added the allele frequency obtained from the gnomAD database for each variant in *SLF2* and *SMC5* to Figure 1a. A '-' in the table indicates that the variant was not present in the gnomAD database. We have also indicated the Polyphen-2 scores of predicted pathogenicity for the missense variants identified in *SLF2* and *SMC5*.

Whilst we have not functionally examined every variant to the same extent, we present multiple lines of evidence in our manuscript that the mutations identified in either *SLF2* or *SMC5* are pathogenic (listed below), and with the above additional information, we believe that we have provided sufficient evidence to demonstrate the disease-causing nature of the identified gene variants in *SLF2* and *SMC5*.

1. Cell lines from four unrelated patients with biallelic variants in *SLF2* display a significant reduction in the expression of *SLF2* protein (Figure 2a), and the ability of *RAD18* to bind *SMC6* under conditions of replication stress is compromised in these cells (Figure 2c).
2. Six of the seven patient-associated mutations in *SLF2* and all three patient-associated mutations in *SMC5* affect the relocalisation of the mutated protein to sites of DNA damage (Supplementary Figure 4).
3. One of the seven patient-associated mutations in *SLF2*-P2 (c.3486G>C; p.Gln1162His) affects both gene splicing and protein stability (Figure 2a and Supplementary Figure 3).
4. Depletion/loss of *SLF2* and *SMC5* gives rise to microcephaly in zebrafish (similar to that observed in patients with mutations in *SLF2* and *SMC5*) (Figure 3), which, in the case of the *SMC5*, can be complemented following re-expression of WT or a polymorphic variant of *SMC5* (p.Arg733Gln) but not the three identified patient-associated mutations (p.Arg372del, p.Arg425Ter and p.His990Asp) (Supplementary Figure 9).

5. Loss of R372 in *SMC5* compromises the binding of *SMC5* to *NSMCE2* (Figure 2e and 2f).
6. Two *SLF2* and three *SMC5* mutant patient-derived cell lines exhibit similar replication abnormalities (Figure 5 and Supplementary Figure 11), which can be complemented by adding back either WT *SLF2* or WT *SMC5* (Figure 5).
7. Two *SLF2* and three *SMC5* mutant patient-derived cell lines exhibit increased levels of spontaneous genome instability (Figures 6-8) which can be complemented by adding back either WT *SLF2* or WT *SMC5* (Figures 6-8).

Based on this, we believe that we have provided a large amount of data independently supporting the pathogenicity of the identified mutations in *SLF2* and *SMC5*.

- The way the first patient is separated out from the others does not make sense – this is a cohort describe it as a cohort.

Response: We separated patient P1 because this was the first patient identified with biallelic mutations in *SLF2*, which then prompted us to search for clinicians/scientists with additional patients with biallelic mutations in either *SLF1*, *SLF2*, *SMC5* and *SMC6*. A cohort of patients was subsequently accrued over time following matches for these genes on GeneMatcher. However, we have altered the text to describe all the *SLF2* mutant patients as a single cohort.

- No note of segregation in P1, was this done. P5 is missing segregation given not available, was CNV analysis done to confirm there is no deletion in trans given its homozygous?

Response: Segregation analysis was not carried out for patient P1 since parental material was not available for exome sequencing. CGH array analysis was carried out for patient P5, which confirmed the patient to be homozygous for the identified *SLF2* gene mutation rather than having a deletion of one of the *SLF2* alleles. A note has been added to the text to indicate this.

- Protein nomenclature should include brackets as per HGVS. This is missing in many figures and some places in the text. Please correct.

Response: We apologies for this oversight. We have corrected the protein nomenclature.

- Figure 2. All of these Western blots are over-exposed and not possible to draw conclusions on. Do the authors have better exposures from which to draw conclusions? Also a loading control not in this pathway should be shown or total protein staining.

Response: We respectfully disagree that it is not possible to draw conclusions from Western blots shown in Figure 2. It is clear from Figure 2a that cell extracts derived from patient cell lines *SLF2*-P1, *SLF2*-P2, *SLF2*-P3 and *SLF2*-P4 exhibit a significant reduction in the expression of *SLF2* but not *ATR*, *SMC6*, *SMC5* and *RAD18*. Furthermore, it is evident that cell extracts derived from patient cell lines *SMC*-P7 exhibit an approximate 50% reduction in the expression of *SMC5*, which is consistent with the fact that this patient has one *SMC5* allele containing a truncating mutation

and the other, an expressed inframe deletion. Lastly, it is evident from our Western blots that the p.(His990Asp) mutation present in patient cell lines SMC5-P8 and SMC5-P9-1 does not affect the stability of SMC5.

The Western blots presented in Figure 2c,2d and 2f are co-immunoprecipitations and as such the amount of immunoprecipitated protein and the protein present in the input lanes are always far higher than the co-immunoprecipitated protein. As such these lanes are often overexposed so that the co-immunoprecipitated protein can be visualised. However, it is the amount of co-immunoprecipitated protein that is the important aspect of these figures. Our data clearly shows that RAD18 does not co-immunoprecipitate with SMC6 in the cell lines from the SLF2 mutant patients, whereas it does to various degrees in the SMC5 mutant cell lines (depending on the mutation).

Lastly, we believe that proteins that function within the SMC5/6 pathway are the most relevant controls for protein loading. Whilst many groups often use γ -tubulin or β -actin or even ponceau S staining of the nitrocellulose filter to control for protein loading, we do not feel that these are always an appropriate control since γ -tubulin or β -actin are present in cells at a far higher concentration when compared to proteins present within the SMC5/6 pathway. As such, small variations in protein loading can be masked when using γ -tubulin or β -actin or ponceau S to control for protein loading. However, we would like to point out that the Western blots presented in figure 2a and 2b have a cross-reactive band for the SLF2 antibody that can be used for protein loading. Failing that, ATR levels have also been examined in figure 2a and 2b, and whilst ATR is a known replication stress responsive protein, it is not a component of the SMC5/6 complex.

REVIEWERS' COMMENTS

Reviewer #1 (Remarks to the Author):

My concerns have been addressed and the manuscript is much improved by the incorporation of the reviewers comments. It is an interesting and extensive analysis, which deserves publication in Nature Communications.

Reviewer #2 (Remarks to the Author):

The authors have done a great job in addressing all the reviewers comments for a study already very accomplished. Overall, this is a very interesting work that was further improved by the revision and deserves to be published in Nature Communications.

Reviewer #1 (Remarks to the Author):

My concerns have been addressed and the manuscript is much improved by the incorporation of the reviewers comments. It is an interesting and extensive analysis, which deserves publication in Nature Communications.

Response: We thank the reviewer for their positive comments.

Reviewer #2 (Remarks to the Author):

The authors have done a great job in addressing all the reviewers comments for a study already very accomplished. Overall, this is a very interesting work that was further improved by the revision and deserves to be published in Nature Communications.

Response: We thank the reviewer for their positive comments.